# MANY-BODY APPROXIMATION FOR NON-NEGATIVE TENSORS

## ABSTRACT

We propose a nonnegative tensor decomposition with focusing on the relationship between the modes of tensors. Traditional decomposition methods assume low-rankness in the representation, resulting in difficulties in global optimization and target rank selection. To address these problems, we present an alternative way to decompose tensors, a *many-body approximation* for tensors, based on an information geometric formulation. A tensor is treated via an energy-based model, where the tensor and its mode correspond to a probability distribution and a random variable, respectively, and many-body approximation is performed on it by taking the interaction between variables into account. Our model can be globally optimized in polynomial time in terms of the KL divergence minimization, which is empirically faster than low-rank approximations keeping comparable reconstruction error. Furthermore, we visualize interactions between modes as *tensor networks* and reveal a nontrivial relationship between many-body approximation and low-rank approximation.

## 1 INTRODUCTION

Tensors are generalization of vectors and matrices. Data in various fields such as neuroscience (Erol & Hunyadi, 2022), bioinformatics (Luo et al., 2017), signal processing (Cichocki et al., 2015), and computer vision (Panagakis et al., 2021) are often stored in the form of tensors, and features are extracted from them. *Tensor decomposition* and its non-negative version (Shashua & Hazan, 2005) are popular methods that extract features by approximating tensors by the sum of products of smaller tensors. These smaller tensors are often called *factors*. It usually tries to minimize the difference between the tensor reconstructed from obtained factors and an original tensor, called the reconstruction error.

In most of tensor decomposition approaches, a *low-rank structure* is typically assumed, where a given tensor is approximated by a linear combination of a small number of bases. Such decomposition requires the following two information. First, it requires the structure, which specifies the type of decomposition such as CP decomposition (Hitchcock, 1927) and Tucker decomposition (Tucker, 1966). In recent years, *tensor networks* (Cichocki et al., 2016) have been introduced, which can intuitively and flexibly design the structure including tensor train decomposition (Oseledets, 2011), tensor ring decomposition (Zhao et al., 2016), and tensor tree decomposition (Murg et al., 2010). Second, it requires the rank value, the number of bases used in the decomposition. Since larger ranks increase the capability of the model while increasing the computational cost, the user is required to find the appropriate rank in this tradeoff problem. Since the above tensor decomposition via minimization of the reconstruction error is non-convex, which causes initial value dependence (Kolda & Bader, 2009, Chapter 3), the problem of finding an appropriate setting of the low-rank structure is highly nontrivial in practice as it is hard to locate the cause if the decomposition does not perform well. As a result, to find proper structure and rank, the user often needs to perform decomposition multiple times with various settings, which is time and memory consuming.

Instead of the low-rank structure that has been the focus of attention in the past, in this paper, we propose a novel formulation of tensor decomposition, called *many-body approximation*, that focuses on the relationship among modes of tensors. We determine the structure of decomposition based on the existence of the interactions between modes. The proposed method requires only the decomposition structure naturally determined by the interactions between the modes and does not

require the rank value, which traditional decomposition methods also require and often suffer to determine.

To describe interactions between modes, we follow the standard strategy in statistical mechanics that uses an energy function $\mathcal{H}(\cdot)$ to treat interactions and considers the corresponding distribution $\exp(\mathcal{H}(\cdot))$. This model is known to be an energy-based model in machine learning, which has been used in Legendre decomposition (Sugiyama et al., 2018; 2016) that decomposes tensors via convex optimization. Technically, it finds factors of a tensor by treating a tensor as a probability distribution and enforcing some of its natural parameters to be zero. We point out that interactions in the energy function $\mathcal{H}(\cdot)$ can be represented using natural parameters of distribution, and we can successfully formulate many-body approximation as a special case of Legendre decomposition by setting some of natural parameters to be zero. The advantage of this approach is that many-body approximation can be also achieved by convex optimization that minimizes the Kullback–Leibler (KL) divergence (Kullback & Leibler, 1951). Our approach, describing interactions between modes using energy functions, is different from existing methods that focus on interactions between mode matrices (Vasilescu & Terzopoulos, 2002; Vasilescu, 2011) or block tensors (Vasilescu et al., 2021).

Furthermore, we introduce a way of representing tensor interactions, which visualizes the presence or absence of interactions between modes. We discuss the correspondence between our representation and the tensor network and point out that an operation called coarse-grained transformation (Levin & Nave, 2007), in which multiple tensors are viewed as a new tensor, reveals unexpected relationship between the proposed method and existing methods such as tensor ring and tensor tree decomposition.

We summarize our contribution as follows:

- By focusing on the interaction between modes of tensors, we introduce an alternative rank-free tensor decomposition, many-body approximation. This decomposition is realized by convex optimization.
- We present a way of describing tensor many-body approximation, interaction representation, a diagram that shows interactions within a tensor. This diagram can be transformed into tensor networks, which tells us the relationship between many-body approximation and existing low-rank approximation.
- We empirically show that many-body approximation is faster than low-rank approximation with competitive reconstruction errors.

## 2 TENSOR MANY-BODY APPROXIMATION

Our proposal, tensor many-body approximation, is based on the formulation of Legendre decomposition for tensors. We first review Legendre decomposition and its optimization in Section 2.1. We introduce interactions between modes and its visual representation to prepare for many-body approximation in Section 2.2. Using interactions between modes, we define many-body approximation in Section 2.3. Finally, we transform the interaction representation into a tensor network and point out the connection between many-body approximation and existing low-rank decomposition methods in Section 2.4.

In the following discussion, we consider $D$-order non-negative tensors whose size is $(I_1, \ldots, I_D)$. We assume the sum of all elements in $\mathcal{P}$ is 1 for simplicity, while this assumption can be eliminated using the general property of Kullback–Leibler (KL) divergence, $\lambda D_{KL}(\mathcal{P}, \mathcal{Q}) = D_{KL}(\lambda \mathcal{P}, \lambda \mathcal{Q})$, for any real number $\lambda$.

### 2.1 REMINDER TO LEGENDRE DECOMPOSITION AND ITS OPTIMIZATION

Legendre decomposition is a method to decompose a non-negative tensor by regarding the tensor as a discrete distribution and representing it with a limited number of parameters. We describe a non-negative tensor $\mathcal{P}$ using natural parameters $\boldsymbol{\theta} = (\theta_{1,\ldots,1}, \ldots, \theta_{I_1,\ldots,I_D})$ and its energy function $\mathcal{H}$ as

$$\mathcal{P}_{i_1,\ldots,i_D} = \exp(\mathcal{H}_{i_1,\ldots,i_D}), \quad \mathcal{H}_{i_1,\ldots,i_D} = \sum_{i_1'=1}^{i_1} \cdots \sum_{i_D'=1}^{i_D} \theta_{i_1',\ldots,i_D'}, \tag{1}$$

where $\theta_{1,\dots,1}$ has a role of normalization. Here it is clear that a tensor corresponds to a distribution whose sample space is its index set; that is, the value of each element is regarded as the probability of realizing the corresponding index (Sugiyama et al., 2017).

As we can see in equation 1, we can uniquely identify tensors from natural parameters $\boldsymbol{\theta}$. We can compute the natural parameter $\boldsymbol{\theta}$ from a given tensor as

$$\theta_{i_1,\dots,i_D} = \sum_{i'_1=1}^{I_1} \cdots \sum_{i'_D=1}^{I_D} \mu_{i_1,\dots,i_D}^{i'_1,\dots i'_D} \log \mathcal{P}_{i'_1,\dots,i'_D} \tag{2}$$

using the Möbius function $\mu : S \times S \to \{-1, 0, +1\}$, where $S$ is the set of indices, defined inductively as follows:

$$\mu_{i_1,\dots,i_D}^{i'_1,\dots,i'_D} = \begin{cases} 1 & \text{if } i_d = i'_d \text{ for all } d \in \{1,\dots,D\}, \\ -\prod_{d=1}^{D}\sum_{j_d=i_d}^{i'_d-1} \mu_{i_1,\dots,i_D}^{j_1,\dots j_D} & \text{else if } i_d \leq i'_d \text{ for all } d \in \{1,\dots,D\}, \\ 0 & \text{otherwise.} \end{cases}$$

The above modelling for non-negative tensors is an instance of the log-linear model on posets (Sugiyama et al., 2017). Since distribution described by equation 1 belongs to the exponential family, we can also identify each tensor by expectation parameters $\boldsymbol{\eta} = (\eta_{1,\dots,1},\dots,\eta_{I_1,\dots,I_D})$ using the Möbius inversion formula as

$$\eta_{i_1,\dots,i_D} = \sum_{i'_1=i_1}^{I_1} \cdots \sum_{i'_D=i_D}^{I_D} \mathcal{P}_{i'_1,\dots,i'_D}, \quad \mathcal{P}_{i_1,\dots,i_d} = \sum_{i'_1=1}^{I_1} \cdots \sum_{i'_D=1}^{I_D} \mu_{i_1,\dots,i_d}^{i'_1,\dots,i'_d} \eta_{i'_1,\dots,i'_d}. \tag{3}$$

See Supplemental Materials for examples of the above calculation. Since distribution is determined by specifying either $\theta$-parameters or $\eta$-parameters, they form two coordinate systems called the $\theta$-coordinate system and the $\eta$-coordinate system, respectively. By using the dual flatness, Legendre decomposition achieves convex optimization as shown in the following.

### 2.1.1 OPTIMIZATION

Legendre decomposition approximates a tensor by setting some $\theta$ values to be zero, which corresponds to dropping some parameters for regularization. Let $B$ be the set of indices of $\theta$ parameters that are not imposed to be 0. Then Legendre decomposition coincides with a projection of a given nonnegative tensor $\mathcal{P}$ onto the subspace $\mathcal{B} = \{\boldsymbol{\theta} \mid \theta_{i_1,\dots,i_D} = 0 \text{ if } (i_1,\dots,i_D) \notin B\}$.

Let us consider projection of a given tensor $\mathcal{P}$ onto $\mathcal{B}$. The space of probability distributions is not a Euclidean space. Therefore, it is necessary to consider geometry of probability distributions, which is studied in information geometry. It is known that a subspace with linear constraints on natural parameters $\theta$ is flat, called $e$-flat (Amari, 2016, Chapter 2). The subspace $\mathcal{B}$ is $e$-flat, meaning that the logarithmic combination, or called $e$-geodesic, $\mathcal{R} \in \{(1-t)\log \mathcal{Q}_1 + t\log \mathcal{Q}_2 - \phi(t) \mid 0 < t < 1\}$ of any two points $\mathcal{Q}_1, \mathcal{Q}_2 \in \mathcal{B}$ is included in the subspace $\mathcal{B}$, where $\phi(t)$ is a normalizer. There is always a unique point $\overline{\mathcal{P}}$ on the $e$-flat subspace that minimizes the KL divergence from any point $\mathcal{P}$.

$$\overline{\mathcal{P}} = \underset{\mathcal{Q};\mathcal{Q}\in\mathcal{B}}{\arg\min}\, D_{KL}(\mathcal{P}, \mathcal{Q}) \tag{4}$$

This projection is called the $m$-projection. The $m$-projection onto a $e$-flat subspace is a convex optimization. We define two vectors $\boldsymbol{\theta}^B = (\theta_b)_{b\in B}$ and $\boldsymbol{\eta}^B = (\eta_b)_{b\in B}$. We write as $|\mathcal{B}|$ the number of elements in these vectors since it is equal to the cardinality of $\mathcal{B}$. The derivative of the KL divergence and the Hessian matrix $G \in \mathbb{R}^{|\mathcal{B}|\times|\mathcal{B}|}$ are given as

$$\frac{\partial}{\partial \boldsymbol{\theta}^B} D_{KL}(\mathcal{P}, \mathcal{Q}) = \boldsymbol{\eta}^B - \hat{\boldsymbol{\eta}}^B, \quad G_{u,v} = \eta_{\max(i_1,j_1),\dots,\max(i_D,j_D)} - \eta_{i_1,\dots,i_D}\eta_{j_1,\dots,j_D} \tag{5}$$

where $\boldsymbol{\eta}^B$ and $\hat{\boldsymbol{\eta}}^B$ are the expectation parameters of $\mathcal{Q}$ and $\mathcal{P}$, respectively, and $u = (i_1,\dots,i_D), v = (j_1,\dots,j_D) \in B$. This matrix $G$ is also known as the negative Fisher information matrix. Using gradient descent with second-order derivative, we can update $\boldsymbol{\theta}^B$ in each iteration $t$ as

$$\boldsymbol{\theta}_{t+1}^B = \boldsymbol{\theta}_t^B - G^{-1}(\boldsymbol{\eta}_t^B - \hat{\boldsymbol{\eta}}^B) \tag{6}$$

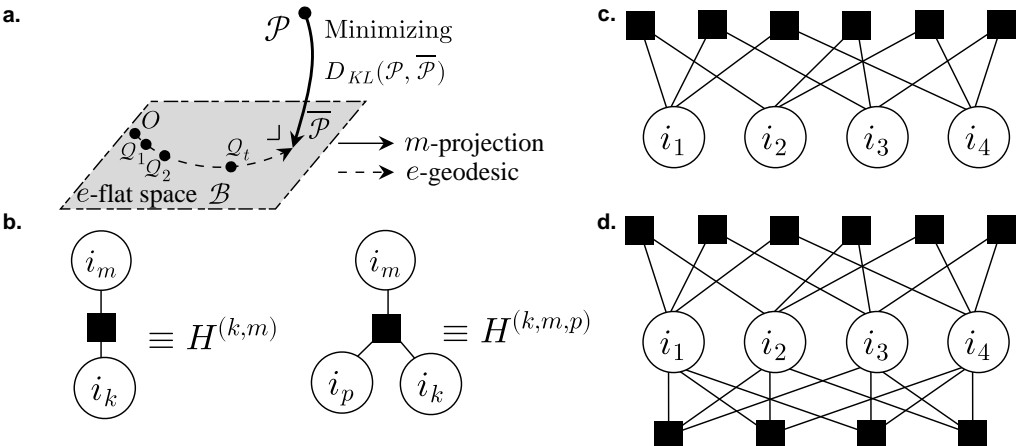

Figure 1: (a) An illustration of optimization of Legendre decomposition. Interaction representations corresponding to (c) equation 10 and (d) equation 11.

The distribution $\mathcal{Q}_{t+1}$ is calculated from the updated natural parameters $\boldsymbol{\theta}_{t+1}$. This step finds a point $\mathcal{Q}_{t+1} \in \mathcal{B}$ that is closer to the destination $\overline{\mathcal{P}}$ along with the $e$-geodesic from $\mathcal{Q}_t$ to $\overline{\mathcal{P}}$. We can also calculate the expected value parameters $\boldsymbol{\eta}_{t+1}$ from the distribution. By repeating this process until convergence, we can always find the globally optimal solution satisfying equation 4. This procedure is illustrated in Figure. 1(a). See Supplemental Materials for more detail on the optimization.

## 2.2 INTERACTION AND ITS REPRESENTATION OF TENSORS

In this subsection, we introduce interactions between modes and its visual representation to prepare for many-body approximation. The following discussion enables us to intuitively describe relationships between modes and formulate our novel rank-free tensor decomposition.

First we introduce $n$-body parameters, which is a generalized concept of one-body and two-body parameters in (Ghalamkari & Sugiyama, 2022). Let $n$ of a $n$-body parameter be the number of non-one indices; for example, $\theta_{1,2,1,1}$ is a one-body parameter, $\theta_{4,3,1,1}$ is a two-body parameter and $\theta_{1,2,4,3}$ is a three-body parameter. We also use the following notation for $n$-body parameters:

$$\theta_{i_k}^{(k)} = \theta_{1,\ldots,1,i_k,1,\ldots,1}, \quad \theta_{i_k,i_m}^{(k,m)} = \theta_{1,\ldots,1,i_k,1,\ldots,1,i_m,1,\ldots,1}, \quad \theta_{i_k,i_m,i_p}^{(k,m,p)} = \theta_{1,\ldots,i_k,\ldots,i_m,\ldots,i_p,\ldots,1},$$

for $n = 1, 2,$ and $3$, respectively. We write the energy function $\mathcal{H}$ with $n$-body parameters as

$$\mathcal{H}_{i_1,\cdots,i_D} = H_0 + \sum_{k=1}^{D} H_{i_k}^{(k)} + \sum_{m=1}^{k-1}\sum_{k=1}^{D} H_{i_k,i_m}^{(k,m)} + \sum_{p=1}^{m-1}\sum_{m=1}^{k-1}\sum_{k=1}^{D} H_{i_k,i_m,i_p}^{(k,m,p)} + \cdots + H_{i_1,\ldots,i_D}^{(1,\ldots,D)} \quad (8)$$

where the $n$-th order energy is introduced as

$$H_{i_{l_1},\ldots,i_{l_n}}^{(l_1,\ldots,l_n)} = \sum_{i'_{l_1}=2}^{i_{l_1}} \cdots \sum_{i'_{l_n}=2}^{i_{l_n}} \theta_{i'_{l_1},\ldots,i'_{l_n}}^{(l_1,\ldots,l_n)}. \quad (9)$$

For simplicity, we suppose that $1 \leq l_1 < l_2 < \cdots < l_n \leq D$ holds. We set $H_0 = \theta_{1,\ldots,1}$. We say that an $n$-body interaction exists between modes $l_1, \ldots, l_n$ if there are indices $i_{l_1}, \ldots, i_{l_n}$ satisfying $H_{i_{l_1},\ldots,i_{l_n}}^{(l_1,\ldots,l_n)} \neq 0$.

The first term $H_0$ in equation 8 is called the normalized factor or the partition function. The terms $H^{(k)}$ are called bias in machine learning and magnetic field or self-energy in statistical physics. The terms $H^{(k,m)}$ are called the weight of the Boltzmann machine in machine learning and two-body interaction or electron-electron interaction in physics.

To visualize the existence of interactions within a tensor, we newly introduce a diagram called *interaction representation*, which is inspired by factor graphs in graphical modelling (Bishop &

Nasrabadi, 2006, Chapter 8). The graphical representation of the product of tensors is widely known as tensor networks. However, displaying the relations between the modes of a tensor as a factor graph is our novel approach. We represent the $n$-body interaction as a black square, $\blacksquare$, connected with $n$ modes. We describe examples of the two-body interaction between modes $(k, m)$ and the three-body interaction among modes $(k, m, p)$ in Figure 1(b). Combining these interactions, the energy function including all two-body interactions is shown in Figure 1(c), and the energy function including all two-body and three-body interactions is shown in Figure 1(d) for $D = 4$.

This visualization allows us to intuitively understand the relationship between modes of tensors. For simplicity, we abbreviate one-body interactions in the diagrams, while we always assume them. Once interaction representation is given, we can determine the corresponding decomposition of tensors.

In the following section, we reduce some of $n$-body interactions, that is, $H_{i_{l_1},\ldots,i_{l_n}}^{(l_1,\ldots,l_n)} = 0$, by fixing each parameter $\theta_{i_{l_1},\ldots,i_{l_n}}^{(l_1,\ldots,l_n)} = 0$ for all indices $(i_{l_1}, \ldots, i_{l_n}) \in \{2, \ldots, I_{l_1}\} \times \cdots \times \{2, \ldots, I_{l_n}\}$.

## 2.3 MANY-BODY APPROXIMATION

Our proposed method, tensor many-body approximation, approximate a given tensor with assuming the existence of dominant interactions between the modes of the tensor and ignoring other interactions. Since this operation can be understood as setting some natural parameters of the distribution to be zero, it can be achieved by convex optimization through the theory of Legendre decomposition. As we see below, approximated tensors are represented without the summation symbol $\sum$. This property is different from existing low-rank approximations except for rank-1 approximation.

As an example, we consider two types of approximations of a nonnegative tensor $\mathcal{P}$ by tensors represented in Figure 1(c), (d). If all energies greater than 2nd-order or those than 3rd-order in equation 8 are ignored, that is, $H_{i_{l_1},\ldots,i_{l_n}}^{(l_1,\ldots,l_n)} = 0$ for $n > 2$ or $n > 3$, $\mathcal{P}$ is approximated as follows:

$$\mathcal{P}_{i_1,i_2,i_3,i_4} \simeq \mathcal{P}_{i_1,i_2,i_3,i_4}^{\leq 2} = X_{i_1,i_2}^{(1,2)} X_{i_1,i_3}^{(1,3)} X_{i_1,i_4}^{(1,4)} X_{i_2,i_3}^{(2,3)} X_{i_2,i_4}^{(2,4)} X_{i_3,i_4}^{(3,4)}, \tag{10}$$

$$\mathcal{P}_{i_1,i_2,i_3,i_4} \simeq \mathcal{P}_{i_1,i_2,i_3,i_4}^{\leq 3} = \chi_{i_1,i_2,i_3}^{(1,2,3)} \chi_{i_1,i_2,i_4}^{(1,2,4)} \chi_{i_1,i_3,i_4}^{(1,3,4)} \chi_{i_2,i_3,i_4}^{(2,3,4)}, \tag{11}$$

where each factor on the right-hand side is represented as

$$X_{i_k,i_m}^{(k,m)} = \frac{1}{\sqrt[6]{Z}} \exp\left(\frac{1}{3} H_{i_k}^{(k)} + H_{i_k,i_m}^{(k,m)} + \frac{1}{3} H_{i_m}^{(m)}\right),$$

$$\chi_{i_k,i_m,i_p}^{(k,m,p)} = \frac{1}{\sqrt[4]{Z}} \exp\left(\frac{H_{i_k}^{(k)} + H_{i_m}^{(m)} + H_{i_p}^{(p)}}{3} + \frac{1}{2} H_{i_k,i_m}^{(k,m)} + \frac{1}{2} H_{i_m,i_p}^{(m,p)} + \frac{1}{2} H_{i_k,i_p}^{(k,p)} + H_{i_k,i_m,i_p}^{(k,m,p)}\right).$$

The partition function, or the normalization factor, is given as $Z = \exp\left(-\theta_{1,1,1,1}\right)$, which do not depend on indices $(i_1, i_2, i_3, i_4)$. When the tensor $\mathcal{P}$ is approximated by $\mathcal{P}^{\leq m}$, the set $B$ contains only indices of $n(\leq m)$-body parameters.

In the above discussion, we consider many-body approximation with all $n$-body parameters, while our formulation allows us to use only a part of $n$-body interactions as shown in the following. We consider the situation where only one-body interaction and two-body interaction between $(k, k+1)$ exist for all $k \in \{1, \ldots, D\}$ ($D + 1$ implies 1 for simplicity). Figure 2(a) shows the interaction representation of the approximated tensor. As we can confirm by substituting 0 for $H_{i_k,i_l}^{(k,l)}$ if $l \neq k+1$, we can describe the approximated tensor as

$$\mathcal{P}_{i_1,\ldots,i_D} \simeq \mathcal{P}_{i_1,\ldots,i_D}^{\text{cyc}} = X_{i_1,i_2}^{(1)} X_{i_2,i_3}^{(2)} \ldots X_{i_D,i_1}^{(D)} \tag{12}$$

where

$$X_{i_k,i_{k+1}}^{(k)} = \frac{1}{\sqrt[D]{Z}} \exp\left(\frac{1}{2} H_{i_k}^{(k)} + H_{i_k,i_{k+1}}^{(k,k+1)} + \frac{1}{2} H_{i_{k+1}}^{(k+1)}\right). \tag{13}$$

with normalization factor $Z = \exp\left(-\theta_{1,\ldots,1}\right)$. When the tensor $\mathcal{P}$ is approximated by $\mathcal{P}^{\text{cyc}}$, the set $B$ contains only all one-body parameters and two-body parameters $\theta_{i_d,i_{d+1}}^{(d,d+1)}$ for $d \in \{1, 2, \ldots, D\}$. We call this approximation *cyclic two-body approximation* since the order of indices in equation 12 is cyclic. We show the connection between cyclic two-body approximation and existing tensor ring decomposition in the following subsection.

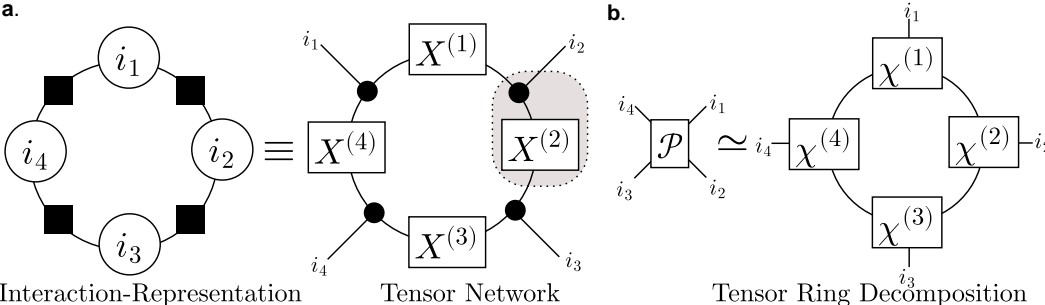

Figure 2: (a) Interaction representation of an example of cyclic two-body approximation and its transformed tensor network for $D = 4$. (b) Tensor network of tensor ring decomposition.

## 2.4 CONNECTION TO TENSOR NETWORK

Tensor interaction representation is a diagram that focuses on the relationship between modes. Tensor networks, which are well known as diagrams that focus on factors after decomposition, represent a tensor as an undirected graph, whose nodes correspond to matrices or tensors and edges are the modes of summation in tensor products (Cichocki et al., 2016).

Our tensor interaction representation has a tight connection to tensor networks, and we can convert a tensor interaction representation to a tensor network. For the conversion, we use a hyper-diagonal tensor $\Omega$, that is defined as $\Omega_{ijk} = \delta_{ij}\delta_{jk}\delta_{ki}$, where $\delta_{ij} = 1$ if $i = j$ and 0 otherwise. The tensor $\Omega$ is often represented by $\bullet$ in tensor networks. In the community of tensor network, the tensor $\Omega$ appears in the CNOT gate and a special case of Z spider (Nielsen & Chuang, 2010). The tensor network in Figure 2(a) represents the following formula

$$\prod_{d=1}^{D} \left( \sum_{j_d} \sum_{l_d} X_{l_d,j_{d+1}}^{(d)} \Omega_{j_{d+1},i_{d+1},l_{d+1}} \right), \tag{14}$$

where $j_{D+1} = j_1, i_{D+1} = i_1, l_{D+1} = l_1$. Substituting the definition of $\Omega$ in equation 14, we realize that the tensor network corresponds to equation 12.

We point out that the tensor network representation of cyclic two-body approximation is similar to the tensor network of the tensor ring decomposition. The tensor ring decomposition is an extension of the tensor train decomposition, and its representation is shown in Figure 2(b) using a tensor network. In fact, if we consider the region enclosed by the dotted line in the tensor network as a new tensor, the tensor network of the cyclic two-body approximation coincides with the tensor network of the tensor ring decomposition(See more details in the Supplemental Materials). This operation, in which multiple tensors are regarded as a new tensor in a tensor network, is called renormalization or coarse-graining transformation (Evenbly & Vidal, 2015).

**Comparing the number of parameters** The number of elements of an input tensor is $I_1 \times I_2 \times \cdots \times I_D$. After the cyclic two-body approximation, the number of parameters is given as

$$|\mathcal{B}| = 1 + \sum_{d=1}^{D}(I_d - 1) + \sum_{d=1}^{D}(I_d - 1)(I_{d+1} - 1) \tag{15}$$

where we assume $I_{D+1} = I_1$. The first term is for a normalizer, the second is the number of one-body parameters, and the final term is the number of two-body parameters. In contrast, in the tensor ring decomposition with the target rank $(R_1, \ldots, R_D)$, the number of parameters is given as $|\mathcal{R}| = \sum_{k=1}^{D} R_k I_k R_{k+1}$. The ratio of the number of parameters of these two methods $|\mathcal{B}|/|\mathcal{R}|$ is proportional to $I/R^2$ if we assume $R_d = R$ and $I_d = I$ for all $d \in \{1, \ldots, D\}$ for simplicity. Therefore, when the target rank is small and the size of the input tensor is large, the proposed method has more parameters than the tensor ring decomposition.

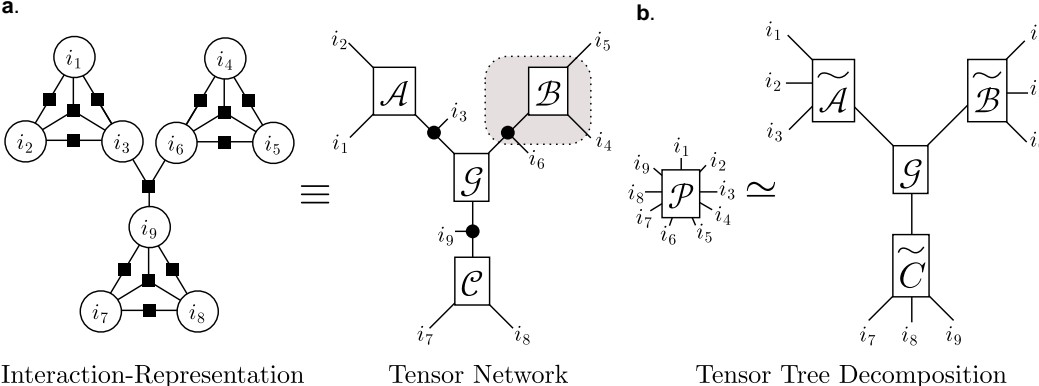

Figure 3: (a) Interaction representation corresponding to equation 16 and its transformed tensor network for $D = 9$. (b) Tensor network of a variant of tensor tree decomposition.

### 2.4.1 OTHER EXAMPLE OF MANY-BODY APPROXIMATION AND ITS TENSOR NETWORK

In the same way, we can find a correspondence between another example of many-body approximation and the existing low-rank approximation. For $D = 9$, we consider three-body and two-body interactions among $(i_1, i_2, i_3)$, $(i_4, i_5, i_6)$, and $(i_7, i_8, i_9)$ and three-body approximation among $(i_3, i_6, i_9)$. We provide the interaction representation of the target energy function in Figure 3(a). In this approximation, the decomposed tensor can be described as

$$\mathcal{P}_{i_1,\ldots,i_9} = \mathcal{A}_{i_1,i_2,i_3}\mathcal{B}_{i_4,i_5,i_6}\mathcal{C}_{i_7,i_8,i_9}\mathcal{G}_{i_3,i_6,i_9}. \tag{16}$$

In the same way in the case of the cyclic two-body approximation, we can convert the interaction representation to a tensor network, as described in Figure 3(a). A tensor network of tensor tree decomposition in Figure 3(b) emerges when the region enclosed by the dotted line is replaced with a new tensor (shown with tilde) in Figure 3(a). Such tensor tree decomposition is used in generative modeling (Cheng et al., 2019), computational chemistry (Murg et al., 2015) and quantum many-body physics (Shi et al., 2006).

As we have seen above, by transforming tensor interaction representation to tensor networks and applying coarse-graining, we can reveal the relationship between tensor many-body approximations and low-rank approximations.

### 2.5 MANY-BODY APPROXIMATION AS GENERALIZATION OF MEAN-FIELD APPROXIMATION

It has been already pointed out that any tensor $\mathcal{P}$ can be represented by vectors $\boldsymbol{x}^{(d)} \in \mathbb{R}^{I_d}$ for $d \in \{1, \ldots, D\}$ as $\mathcal{P}_{i_1,\ldots,i_D} = x_{i_1}^{(1)} x_{i_2}^{(2)} \ldots x_{i_D}^{(D)}$ if and only if all $n(\geq 2)$-body $\theta$-parameters are $0$ (Ghalamkari & Sugiyama, 2021). The right-hand side is equal to the Kronecker product of $D$ vectors $\boldsymbol{x}^{(1)}, \ldots, \boldsymbol{x}^{(D)}$, and therefore this approximation is equivalent to the rank-1 approximation since the rank of the tensor that can be represented by the Kronecker product is always 1, which is also known to correspond to mean-field approximation. In this study, we propose many-body approximation by relaxing the condition for the mean-field approximation that ignores $n(\geq 2)$-body interactions. Therefore many-body approximation is generalization of rank-1 approximation and mean-field approximation.

### 2.6 COMPUTATIONAL COMPLEXITY

We analyze the computational complexity of many-body approximation. In many-body approximation, the overall complexity is dominated by the update of $\theta$, which includes matrix inversion of $G$. The complexity of computing the inverse of an $n \times n$ matrix is $O(n^3)$. Therefore, the computational complexity of many-body approximation is $\mathcal{O}(\gamma|\mathcal{B}|^3)$, where $\gamma$ is the number of iterations.

This complexity can be reduced if we reshape tensors so that the size of each mode becomes small. For example, let us consider a 3-order tensor whose size is $(J^2, J^2, J^2)$ and its cyclic two-

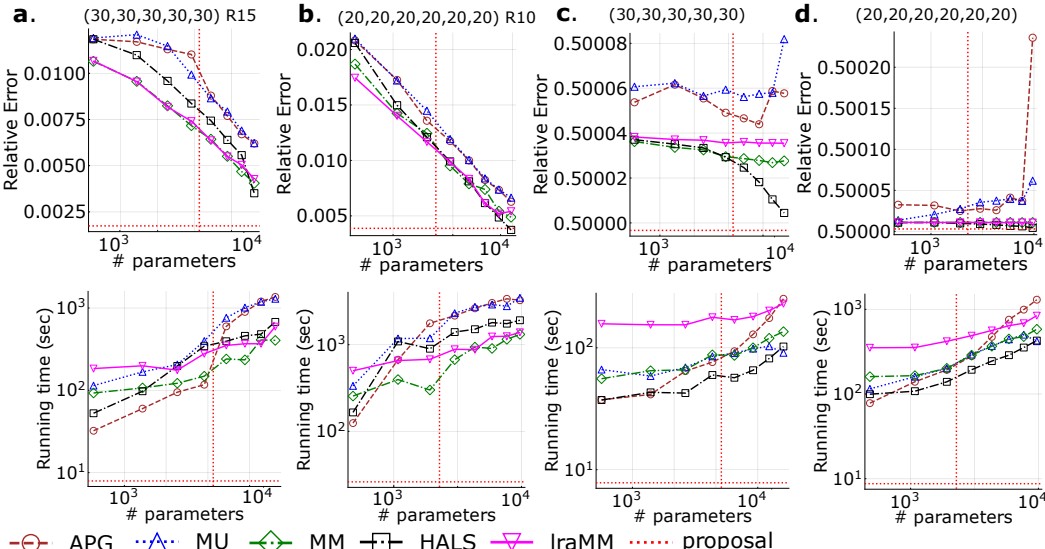

Figure 4: (a)(b) Results for low ring rank tensor. (c)(d) Results for tensors sampled from uniform distribution. The vertical red dotted line is $|\mathcal{B}|$ (See equation 15).

body approximation. In this case, the time complexity is $\mathcal{O}(\gamma J^{12})$ since it holds that $|\mathcal{B}| \propto J^4$ (See equation 15). In contrast, if we reshape the input tensor to a 6-order tensor whose size is $(J, J, J, J, J, J)$, the time complexity becomes $\mathcal{O}(\gamma J^6)$ since it holds that $|\mathcal{B}| \propto J^2$.

This technique of reshaping a tensor into a larger-order tensor is used practically not only in the proposed method but also in various methods based on tensor networks, such as tensor ring decomposition (Malik & Becker, 2021).

## 3 EXPERIMENTS

As seen in Section 2.4, many-body approximation has a close connection to low-rank approximation. For example, in a tensor ring decomposition, if we impose that decomposed factors can be represented as products with hyper-diagonal tensors $\Omega$, this decomposition is equivalent to a cyclic two-body approximation (see Figure 2). Therefore, to examine our conjecture that cyclic two-body approximation is as capable of approximating as tensor ring decomposition, we empirically examine the efficiency and effectiveness of cyclic two-body approximation compared with tensor ring decomposition. As baselines, we use five existing methods of non-negative tensor ring decomposition, NTR-APG, NTR-HALS, NTR-MU, NTR-MM and NTR-lraMM (Yu et al., 2021; 2022). These methods minimize the reconstruction error defined with the Frobenius norm by the gradient method. See Supplemental Materials for implementation detail.

We evaluate the approximation performance by the relative error $\|\mathcal{T} - \overline{\mathcal{T}}\|_F / \|\mathcal{T}\|_F$ for an input tensor $\mathcal{T}$ and a reconstructed tensor $\overline{\mathcal{T}}$ with the Frobenius norms $\|\cdot\|_F$. Since all the existing methods are based on nonconvex optimization, we plot the best score (minimum relative error) among 5 restarts with random initialization. In contrast, the score of our method is obtained by a single run as it is convex optimization and such restarts are fundamentally unnecessary. We compare the total running time of them.

**Synthetic data**  We performed experiments on four synthetic datasets. The first two are synthetic data with low tensor ring rank. This setting is often used in evaluation of tensor ring decomposition. We create $D$ core tensors of size $R \times I \times R$ by sampling from uniform distribution. Then a tensor with the size $I^D$ and the tensor ring rank $(R, \ldots, R)$ is obtained by the product of these $D$ tensors. Results for $R = 15, D = 5, I = 30$ are shown in Figure 4(a)., and those for $R = 10, D = 6, I = 20$ in Figure 4(b). Relative error and computation time are plotted with gradually increasing the target rank of the tensor ring decomposition, which is compared to the score of our method, plotted as the

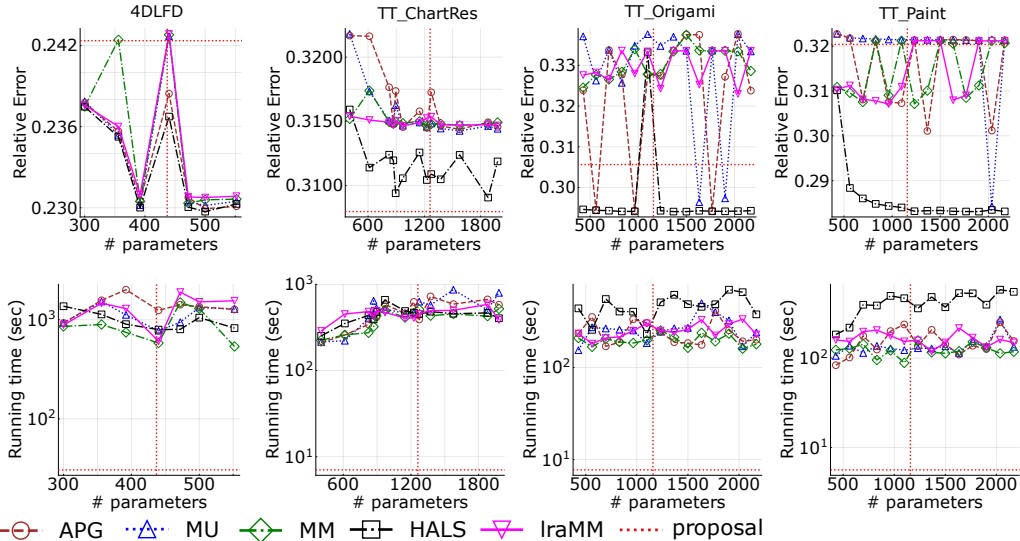

Figure 5: Experimental results for real datasets. The vertical red dotted line is $|\mathcal{B}|$ (See equation 15).

cross point of horizontal and vertical red dotted lines. Please note that our method does not have the concept of the rank, thus the score of our method is invariant to changes of the target rank unlike other methods. If the cross point of red dotted lines is lower than other lines, the proposed method is better than other methods.

In addition to the above case in which we assumed the low-rankness, we also generated synthetic datasets without such an assumption. We created a tensor of size $30^5$ and a tensor of size $20^5$ by sampling from uniform distribution and performed the same experiment. Results are shown in Figure 4(c) and Figure 4(d). In all experiments, the proposed method is superior to comparison partners in both efficiency and effectiveness. It should be noted that the relative error of the proposed method is smaller even when the target rank of the tensor ring decomposition is large and the number of parameters is several times larger than the proposed method.

**Real data**   Next, we evaluate our method on real data. `4DLFD` is a 9-order tensor, which is produced from 4D Light Field Dataset (Honauer et al., 2016; Gortler et al., 1996; Levoy & Hanrahan, 1996). `TT_ChartRes`, `TT_Origami` and `TT_Paint` are 7-order tensors, which is produced from TokyoTech Hyperspectral Image Dataset (Monno et al., 2015; 2017). Each tensor has been reshaped to reduce the computational complexity. See the dataset details in the Supplemental Materials. The proposed method is always faster than baselines with keeping the competitive relative errors. In baseline methods, a slight change of the target rank can induce a significant increase of the reconstruction error due to the nonconvex nature of them.

## 4 CONCLUSION

We propose *many-body approximation* for tensors, which decomposes tensors with focusing on the relationship between modes represented by an energy-based model. It approximates tensors by ignoring the energy corresponding to some interactions, which can be viewed as generalization of mean-field approximation that considers only one-body interactions. Our novel formulation enables us to achieve convex optimization of the model, while the existing approaches based on the low-rank structure are non-convex. Furthermore, we introduce a way of visualize interactions between modes, called interaction representation, to see activated interactions between modes. We have established transformation between our representation and tensor networks, which reveals the nontrivial connection between many-body approximation and the classical tensor low-rank tensor decomposition.

## REPRODUCIBILITY STATEMENT

We provide implementation and dataset details in Supplemental Material. We provide both of code for the proposed method and code for comparison methods in Supplemental Material.

## ETHICS STATEMENT

This study is theoretical analysis of tensors and we believe that our theoretical discussion would not have any negative societal impacts.

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

## A  Cyclic two-body approximation and ring decomposition

We can interpret cyclic two-body approximation as tensor ring decomposition with constraints as described below. Non-negative tensor ring decomposition approximates a given tensor $\mathcal{P} \in \mathbb{R}_{\geq 0}^{I_1 \times \cdots \times I_D}$ with $D$ core tensors $\chi^{(1)}, \chi^{(2)}, \ldots, \chi^{(D)}$ with $\chi^{(d)} \in \mathbb{R}_{\geq 0}^{R_{d-1} \times I_d \times R_d}$ for each $d \in \{1, 2, \ldots, D\}$ as

$$\mathcal{P}_{i_1,\ldots,i_D} \simeq \overline{\mathcal{P}}_{i_1,\ldots,i_D} = \sum_{r_1=1}^{R_1} \sum_{r_2=1}^{R_2} \cdots \sum_{r_D=1}^{R_D} \chi^{(1)}_{r_D,i_1,r_1} \chi^{(2)}_{r_1,i_2,r_2} \cdots \chi^{(D)}_{r_{D-1},i_D,r_D} \qquad (18)$$

where $(R_1, \ldots, R_D)$ is called tensor ring rank. The decomposition is described in Figure 2(c). The cyclic two-body approximation also approximates the tensor $\mathcal{P}$ in the form of equation 18, imposing an additional constraint that each core tensor $\chi^{(d)}$ is decomposed as

$$\chi^{(d)}_{r_{d-1},i_d,r_d} = \sum_{m_d=1}^{I_d} X^{(d)}_{r_{d-1},m_d} \Omega_{m_d,i_d,r_d} \qquad (19)$$

for each $d \in \{1, 2, \ldots, D\}$, where $\Omega_{ijk} = \delta_{ij}\delta_{jk}\delta_{ki}$. We assume $r_0 = r_D$ for simplicity. We obtain equation 12 by substituting equation 19 into equation 18.

This constraint enables us to perform convex optimization. From Kronecker's delta $\delta$, $r_d = i_d$ holds in equation 19, thus $\chi^{(d)}$ is a tensor with the size $I_{d-1} \times I_d \times I_d$. Tensor ring rank after the cyclic two-body approximation is $(I_1, \ldots, I_D)$ since the size of core tensors coincides with tensor ring rank.

## B  Implementation detail

We describe the implementation details of methods in the following.

**Proposed method**  Our method is implemented in Julia 1.8. We use a natural gradient method for cyclic two-body approximation. The natural gradient method uses the inverse of the Fisher information matrix to perform second-order optimization in a non-euclidean space. For non-normalized tensors, we conduct the following procedure. First, we compute the total sum of elements of an input tensor. Then, we normalize the tensor. After that, we conduct Legendre decomposition for the normalized tensor. Finally, we get the product of the result of the previous step and the total sum we compute initially. The termination criterion is the same as the original implementation of Legendre Decomposition by Sugiyama et al. (2018), that is, it terminates if $||\boldsymbol{\eta}_t^B - \hat{\boldsymbol{\eta}}^B|| < 10^{-5}$, where $\boldsymbol{\eta}_t^B$ is the expectation parameters on $t$-th step and $\hat{\boldsymbol{\eta}}^B$ is the expectation parameters of an input tensor, which are defined in Section 2.1.1. The overall procedure is described in Algorithm 1. Note that this algorithm is based on Legendre decomposition by Sugiyama et al. (2018).

**Baseline methods**  We implemented baseline methods by translating MATLAB code provided by the authors into Julia code for fair comparison. As we can see from their original papers, NTR-APG, NTR-HALS, NTR-MU, NTR-MM and NTR-lraMM have an inner and outer loop to find a local solution. We repeat the inner loop 100 times. We stop the outer loop when the difference between the relative error of the previous and the current iteration is less than 10e-4. NTR-MM and NTR-lraMM require diagonal parameters matrix $\Xi$. We define $\Xi = \omega I$ where $I$ is an identical matrix and $\omega = 0.1$. The NTR-lraMM method performs low-rank approximation to the matrix obtained by mode expansion of an input tensor. The target rank is set to be 20. This setting is the default setting in the provided code. The initial positions of baseline methods were sampled from uniform distribution on $(0, 1)$.

---

**Algorithm 1:** Many-body approximation

---

MANYBODYAPPROXIMATION($\mathcal{T}$, $B$)

    $s \leftarrow$ Total sum of $\mathcal{T}$.

    Obtain normalized input tensor $\mathcal{P} \leftarrow \mathcal{T}./s$            // "./" denotes element-wise division

    Compute $\hat{\boldsymbol{\eta}}$ of $\mathcal{P}$ using equation 3.

    Initialize $\boldsymbol{\theta}_{t=1}^{B}$                              // e.g. $\theta_b = 0$ for all $b \in B$

    $t \leftarrow 1$

    **repeat**

        Compute $\mathcal{Q}_t$ using the current parameter $\boldsymbol{\theta}_t^B$ with equation 1.

        Compute $\boldsymbol{\eta}_t^B$ from $\mathcal{Q}_t$ using equation 3.

        Compute the inverse of the Fisher information matrix $G$ using equation 5.

        $\boldsymbol{\theta}_{t+1}^{B} \leftarrow \boldsymbol{\theta}_t^B - G^{-1}(\boldsymbol{\eta}_t^B - \hat{\boldsymbol{\eta}}^B)$

        $t \leftarrow t + 1$

    **until** $||\boldsymbol{\eta}_t^B - \hat{\boldsymbol{\eta}}^B|| < \epsilon$            // We set $\epsilon = 10^{-5}$ in our implementation;

    $\overline{\mathcal{T}} \leftarrow \mathcal{Q}_t .* s$                  // ".*" denotes element-wise multiplication

    **return** $\overline{\mathcal{T}}$

---

**Environment**   Experiments were conducted on Ubuntu 20.04.1 with a single core of 2.1GHz Intel Xeon CPU Gold 5218 and 128GB of memory.

## C   DATASET DETAIL

We describe the details of each dataset in the following.

**Synthetic Datasets**   For all experiments on synthetic datasets, we change the target ring-rank as $(r, \ldots, r)$ for $r = 2, 3, \ldots, 9$ for baseline methods.

**Real Datasets**   `4DLFD` is originally a $(9, 9, 512, 512, 3)$ tensor, which is produced from 4D Light Field Dataset described in Honauer et al. (2016). Its license is Creative Commons Attribution-NonCommercial-ShareAlike 4.0 International License. We use `dino` images and their depth and disparity map in training scenes. We concatenate them to produce a tensor. We reshaped the tensor as $(6, 8, 6, 8, 6, 8, 6, 8, 12)$. For baseline methods, we chose the target ring-rank as $(2, 3, 2, 2, 2, 2, 2, 2, 2)$, $(2, 3, 2, 2, 3, 2, 2, 3, 2)$, $(2, 2, 2, 2, 2, 2, 2, 2, 5)$, $(2, 5, 2, 2, 5, 2, 2, 2, 2)$, $(2, 2, 2, 2, 2, 2, 2, 2, 7)$, $(2, 2, 2, 2, 3, 2, 2, 2, 7)$, $(2, 2, 2, 2, 2, 2, 2, 2, 9)$. `TT_ChartRes` is originally a $(736, 736, 31)$ tensor, which is produced from TokyoTech 31-band Hyperspectral Image Dataset. We use `ChartRes.mat`. We reshaped the tensor as $(23, 8, 4, 23, 8, 4, 31)$. For baseline methods, we chose the target ring-rank as $(2, 2, 2, 2, 2, 2, 2)$ $(2, 2, 2, 2, 2, 2, 5)$, $(2, 2, 2, 2, 2, 2, 8)$, $(3, 2, 2, 3, 2, 2, 5)$, $(2, 2, 2, 2, 2, 2, 9)$, $(3, 2, 2, 3, 2, 2, 6)$, $(4, 2, 2, 2, 2, 2, 6)$, $(3, 2, 2, 4, 2, 2, 8)$, $(3, 2, 2, 3, 2, 2, 9)$, $(3, 2, 2, 3, 2, 2, 10)$, $(3, 2, 2, 3, 2, 2, 12)$, $(3, 2, 2, 3, 2, 2, 15)$, $(3, 2, 2, 3, 2, 2, 16)$. `TT_Origami` and `TT_Paint` are originally $(512, 512, 59)$ tensors, which are produced from TokyoTech 59-band Hyperspectral Image Dataset. We use `Origami.mat` and `Paint.mat`. In `TT_Origami`, 0.0016% of elements were negative, hence we preprocessed all elements of `TT_Origami` by subtracting $-0.000764$, the smallest value in `TT_Origami`, to make all elements non-negative. We reshaped the tensor as $(8, 8, 8, 8, 8, 8, 59)$. For baseline methods, we chose the target ring-rank as $(2, 2, 2, 2, 2, 2, r)$ for $r = 2, 3, \ldots, 15$. These reshaping reduces the computational complexity as described in Section 2.6 to complete all the experiments in a reasonable time.

## D   PROJECTION THEORY IN INFORMATION GEOMETRY

We explain concepts of information geometry used in this study, including natural parameters, expectation parameters, model flatness, and convexity of optimization. In the following discussion, we consider only discrete probability distributions.

$(\theta, \eta)$**-coordinate and geodesics** In this study, we map a normalized $D$-order non-negative tensor $\mathcal{P} \in \mathbb{R}_{\geq 0}^{I_1 \times \cdots \times I_D}$ to a discrete probability distribution with $D$ random variables. Let $\mathcal{U}$ be the set of discrete probability distributions with $D$ random variables. The entire space $\mathcal{U}$ is a non-Euclidean space with the Fisher information matrix $G$ as the metric. This metric measures the distance between two points. In Euclidean space, the shortest path between two points is a straight line. In a non-Euclidean space, such a shortest path is called a geodesic. In the space $\mathcal{U}$, two kinds of geodesics can be introduced, $e$-geodesics and $m$-geodesics. For two points $\mathcal{P}_1, \mathcal{P}_2 \in \mathcal{U}$, $e$- and $m$-geodesics can be defined as

$$\left\{ \mathcal{R}_t \mid \log \mathcal{R}_t = (1-t) \log \mathcal{P}_1 + t \log \mathcal{P}_2 - \phi(t) \right\}, \quad \left\{ \mathcal{R}_t \mid \mathcal{R}_t = (1-t)\mathcal{P}_1 + t\mathcal{P}_2 \right\},$$

respectively, where $0 \leq t \leq 1$ and $\phi(t)$ is a normalization factor to keep $\mathcal{R}_t$ to be a distribution.

We can parameterize distributions $\mathcal{P} \in \mathcal{U}$ by a parameter called the natural parameter. We have described the relationship between a distribution $\mathcal{P}$ and natural parameter $\boldsymbol{\theta} = (\theta_{1,\ldots,1}, \ldots, \theta_{I_1,\ldots,I_D})$ in equation 1. The natural parameter $\boldsymbol{\theta}$ serves as a coordinate system of $\mathcal{U}$, since any distribution in $\mathcal{U}$ is specified by determining $\boldsymbol{\theta}$. Furthermore, we can also specify a distribution $\mathcal{P}$ by its expectation parameter $\boldsymbol{\eta} = (\eta_{1,\ldots,1}, \ldots, \eta_{I_1,\ldots,I_D})$, which corresponds to expected values of the distribution and an alternative coordinate system of $\mathcal{U}$. The definition of the expectation parameter $\boldsymbol{\eta}$ is described in equation 3. $\theta$-coordinates and $\eta$-coordinates are orthogonal with each other, which means that the Fisher information matrix $G$ has the following property, $G_{u,v} = \partial \eta_u / \partial \theta_v$ and $(G^{-1})_{u,v} = \partial \theta_u / \partial \eta_v$. $e$- and $m$-geodesics can also be described using these parameters as follows.

$$\left\{ \boldsymbol{\theta}^t \mid \boldsymbol{\theta}^t = (1-t)\boldsymbol{\theta}^{\mathcal{P}_1} + t\boldsymbol{\theta}^{\mathcal{P}_2} \right\}, \quad \left\{ \boldsymbol{\eta}^t \mid \boldsymbol{\eta}^t = (1-t)\boldsymbol{\eta}^{\mathcal{P}_1} + t\boldsymbol{\eta}^{\mathcal{P}_2} \right\},$$

where $\boldsymbol{\theta}^{\mathcal{P}}$ and $\boldsymbol{\eta}^{\mathcal{P}}$ are $\theta$- and $\eta$-coordinate of a distribution $\mathcal{P} \in \mathcal{U}$.

**Flatness and projections** A subspace is called $e$-flat when any $e$-geodesic connecting two points in a subspace is included in the subspace. The vertical descent of an $m$-geodesic from a point $\mathcal{P} \in \mathcal{U}$ onto $e$-flat subspace $\mathcal{B}_e$ is called $m$-projection. Similarly, $e$-projection is obtained when we replace all $e$ with $m$ and $m$ with $e$. The flatness of subspaces guarantees the uniqueness of the projection destination. The projection destination $\overline{\mathcal{P}}$ or $\tilde{\mathcal{P}}$ obtained by $m$- or $e$-projection onto $\mathcal{B}_e$ or $\mathcal{B}_m$ minimizes the following KL divergence,

$$\overline{\mathcal{P}} = \underset{\mathcal{Q} \in \mathcal{B}_e}{\arg\min} \, D_{KL}(\mathcal{P}, \mathcal{Q}), \quad \tilde{\mathcal{P}} = \underset{\mathcal{Q} \in \mathcal{B}_m}{\arg\min} \, D_{KL}(\mathcal{Q}, \mathcal{P}).$$

The KL divergence from discrete distributions $\mathcal{P} \in \mathcal{U}$ to $\mathcal{Q} \in \mathcal{U}$ is given as

$$D_{KL}(\mathcal{P}, \mathcal{Q}) = \sum_{i_1=1}^{I_1} \cdots \sum_{i_D=1}^{I_D} \mathcal{P}_{i_1,\ldots,i_D} \log \frac{\mathcal{P}_{i_1,\ldots,i_D}}{\mathcal{Q}_{i_1,\ldots,i_D}}. \tag{20}$$

It is known that a subspace with linear constraints on natural parameters $\theta$ is $e$-flat (Amari, 2016, Chapter 2). The proposed many-body approximation performs $m$-projection onto the subspace $\mathcal{B} \subset \mathcal{U}$ with some natural parameters fixed to be 0. From this linear constraint, we know that $\mathcal{B}$ is $e$-flat. Therefore, the optimal solution of the many-body approximation is always unique. When a space is $e$-flat and $m$-flat at the same time, we say that the space is dually-flat. $\mathcal{U}$ is dually-flat.

**Natural gradient method** $e(m)$-flatness guarantees that cost functions to be optimized in equation 20 are convex. Therefore, $m(e)$-projection onto an $e(m)$-flat subspace can be implemented by a gradient method using a second-order gradient. We call this gradient method the natural gradient method. The Fisher information matrix $G$ appears by second-order differentiation of the KL divergence (see equation 5). We can perform fast optimization using the update formula in equation 6, using the inverse of the Fisher information matrix.

**Examples for Möbius function** In the proposed method, we need to transform the distribution $\mathcal{P} \in \mathbb{R}^{I_1 \times \cdots \times I_D}$ with $\boldsymbol{\theta}$ and $\boldsymbol{\eta}$ using the Möbius function, defined in Section 2.1. We provide examples here. In equation 2, The Möbius function is used to find the natural parameter $\boldsymbol{\theta}$ from a distribution $\mathcal{P}$. For example, if $D = 2, 3$, it holds that

$$\theta_{i_1,i_2} = \log \mathcal{P}_{i_1,i_2} - \log \mathcal{P}_{i_1-1,i_2} - \log \mathcal{P}_{i_1,i_2-1} + \log \mathcal{P}_{i_1-1,i_2-1},$$

$$\theta_{i_1,i_2,i_3} = \log \mathcal{P}_{i_1,i_2,i_3} - \log \mathcal{P}_{i_1-1,i_2,i_3} - \log \mathcal{P}_{i_1,i_2-1,i_3} - \log \mathcal{P}_{i_1,i_2,i_3-1}$$
$$+ \log \mathcal{P}_{i_1-1,i_2-1,i_3} + \log \mathcal{P}_{i_1,i_2-1,i_3-1} + \log \mathcal{P}_{i_1-1,i_2,i_3-1} - \log \mathcal{P}_{i_1-1,i_2-1,i_3-1},$$

where we assume $\mathcal{P}_{0,i_2} = \mathcal{P}_{i_1,0} = 1$ and $\mathcal{P}_{i_1,i_2,0} = \mathcal{P}_{i_1,0,i_3} = \mathcal{P}_{0,i_2,i_3} = 1$. Note that, to identify the value of $\theta_{i_1,\dots,i_d}$, we need only $\mathcal{P}_{i'_1,\dots,i'_d}$ with $(i'_1,\dots,i'_d) \in \{i_1 - 1, i_1\} \times \{i_2 - 1, i_2\} \times \cdots \times \{i_d - 1, i_d\}$. In the same way, using equation 3, we can find the distribution $\mathcal{P}$ by the expectaion parameter $\boldsymbol{\eta}$. For example, if $D = 2, 3$, it holds that

$$\mathcal{P}_{i_1,i_2} = \eta_{i_1,i_2} - \eta_{i_1+1,i_2} - \eta_{i_1,i_2+1} + \eta_{i_1+1,i_2+1},$$
$$\mathcal{P}_{i_1,i_2,i_3} = \eta_{i_1,i_2,i_3} - \eta_{i_1+1,i_2,i_3} - \eta_{i_1,i_2+1,i_3} - \eta_{i_1,i_2,i_3+1}$$
$$+ \eta_{i_1+1,i_2+1,i_3} + \eta_{i_1+1,i_2,i_3+1} + \eta_{i_1,i_2+1,i_3+1} - \eta_{i_1+1,i_2+1,i_3+1},$$

where we assume $\eta_{I_1+1,i_2} = \eta_{i_1,I_2+1} = 0$ and $\eta_{I_1+1,i_2,i_3} = \eta_{i_1,I_2+1,i_3} = \eta_{i_1,i_2,I_3+1} = 0$.

