# OpenReview forum: "Many-Body Approximation for Tensors"
_ICLR.cc/2023/Conference — Submitted to ICLR 2023_

### Official Review · Reviewer_w947 · 2022-10-21

**Confidence:** 4
**Correctness:** 4
**Technical Novelty And Significance:** 4
**Empirical Novelty And Significance:** 3
**Recommendation:** 8

**Clarity, Quality, Novelty And Reproducibility:**

The paper is very clear, and the concept, models and algorithm are original and well supported by theory.

**Strength And Weaknesses:**

Strengths:
-	Original model with nice properties such as its rank-free feature and the convexity of the associated optimization problem.
-	Strong theoretical ground based on Information Theory is provided.
-	Very well written and clear presentation of the model and algorithm

Weaknesses:
-	The model only applies to nonnegative tensors, which limits its applicability to many tensor data problems. Current title: “MANY-BODY APPROXIMATION FOR TENSORS” suggests a general model. A more appropriate title would be “MANY-BODY APPROXIMATION FOR NONNEGATIVE TENSORS”
-	The equivalence between the proposed model (cyclic Many Body) and Tensor Ring (TR) should be explored more deeply. For example, given a cyclic Many Body model, it is shown that it is equivalent to a TR network. However, it is not clear what is the rank of the obtained TR model. Also, is the obtained TR-rank the minimal rank? Is there a way to check the actual rank of the proposed model?
-	The experimental results are rather limited:
1)	They only consider a cyclic 2-order interactions model in the experiments, which is equivalent to a TR network. I think more experiments considering higher order interactions would provide rich insights about the expression power of the new model compared to classical tensor decompositions.
2)	I found the experiments with totally random tensors not very useful because in that case there is no structure on the data tensor itself. The value of this experiment, if any, should be better explained in the paper.
3)	The results on real datasets are not properly discussed. The behavior of the proposed algorithm is not well understood. For example, the error on the 4DLFD dataset is larger than other algorithms without any clear explanation about this result. In general, it seems that HALS always provides the best results in terms of error. Is there any explanation about it?

Minor issues:
-	There is a typo or a problem of visualization in Figure 1. The KL distance should be between P and \bar{P}.
-	In section 2.5, I think the following sentence needs to be fixed: “any tensor P can be represented by …” -> “any tensor P can be approximated by … ”, because in general not any tensor can exactly represented as a rank-1 tensor.


**Summary Of The Paper:**

In this paper, a new tensor decomposition for nonnegative tensors is proposed interpreting the tensor as a probability measure over a multidimensional discrete space given by its indices commonly known as a random field.
The authors used tools from Physics in which an Energy function is defined containing terms that corresponds to potentials associated to first order, second order, etc. interactions between variables in the random field.
In this model, instead of using the concept of rank of the decomposition, the complexity is regulated by the order and map of interactions between variables (indices).
For the optimization of the model to fit it to some data, they applied Information Theory tools so that the optimization problem is reduced to a convex one and a natural gradient-based algorithm is derived to minimize a Kulback-Liebler distance.
The paper demonstrates that, for example, by considering a cyclic 2-order interaction the obtained model is equivalent to a Tensor Ring (TR) network.
Experimental results using only a cyclic 2-order interaction model are included demonstrating the good properties in terms of accuracy and computation time of the proposed learning algorithm.


**Summary Of The Review:**

A novel tensor decomposition for nonnegative tensors is proposed and an efficient algorithm supported by information theory ideas is provided. Although the experimental evaluation could be highly improved, I think the community working on tensor decompositions will appreciate this work because it provides new models that are rank-free and with provable convergent algorithms.

---

> ### Author Response · Authors · 2022-11-19
> **Authors' response to Reviewer w947**
>
> Thank you very much for your positive comments. We provide our point-by-point response to each of your comments in the following.
>
> >The model only applies to nonnegative tensors, which limits its applicability to many tensor data problems. Current title: “MANY-BODY APPROXIMATION FOR TENSORS” suggests a general model. A more appropriate title would be “MANY-BODY APPROXIMATION FOR NONNEGATIVE TENSORS”
>
> Thank you for your suggestion. We modified the title to MANY-BODY APPROXIMATION FOR NONNEGATIVE TENSORS.
>
> >The equivalence between the proposed model (cyclic Many Body) and Tensor Ring (TR) should be explored more deeply. For example, given a cyclic Many Body model, it is shown that it is equivalent to a TR network. However, it is not clear what is the rank of the obtained TR model. Also, is the obtained TR-rank the minimal rank? Is there a way to check the actual rank of the proposed mode?
>
> The cost functions to be optimized are different in tensor ring decomposition and cyclic two-body approximation. However, what we are saying is that the cyclic two-body approximation can be viewed as a tensor ring decomposition with constraints. Tensor ring decomposition decomposes a given tensor into some core tensors $\chi^{(d)}$ for $d=\{1,2\dots,D\}$. If we assume each element in the core tensor can be written as $\chi_{ijk}^{(d)} = \sum_m X_{i,m}\Omega_{m,j,k}$, these cost functions becomes equivalent.
> In tensor ring decomposition, described in Figure 2(b), tensor ring rank $(R_1,\dots,R_D)$ is defined by the size of core tensors $\chi^{(d)}\in\mathbb{R}^{R_{d-1}\times I_d \times R_d}$. The region enclosed by the dotted line in Figure 2(a) corresponds to the core tensor in cyclic two-body approximation. The size of the tensor is $I_{d-1}\times I_d \times I_d$. Then, the tensor ring rank of $\mathcal{P}^{cyc}$ is $(I_1,\dots,I_D)$ at most. We have added a more detailed description of the correspondence between these models in Appendix A.
>
> >The experimental results are rather limited:
> >1) They only consider a cyclic 2-order interactions model in the experiments, which is equivalent to a TR network. I think more experiments considering higher order interactions would provide rich insights about the expression power of the new model compared to classical tensor decomposition.
>
> We agree with this point. However, there are many combination candidates for valid interactions when we consider higher order interactions, and investigating them could be out of the scope of our paper. In this paper we focus on the theoretical side and the cyclic two-body approximation to mainly explore its theoretical properties. The experimental analysis of high-orders many-body approximation is an interesting future work.
>
> >2) I found the experiments with totally random tensors not very useful because in that case there is no structure on the data tensor itself. The value of this experiment, if any, should be better explained in the paper.
>
> This experiment is rather sanity check for the methods.
>
> >3) The results on real datasets are not properly discussed. The behavior of the proposed algorithm is not well understood. For example, the error on the 4DLFD dataset is larger than other algorithms without any clear explanation about this result. In general, it seems that HALS always provides the best results in terms of error. Is there any explanation about it?
>
> They should be due to different inductive biases across methods, while more detailed exploration to give clear explanation is not easy as tensor data are generally complicated. In addition, baseline methods tend to be unstable as they are nonconvex optimization.
> We have added the following sentence.
>
> ``In baseline methods, a slight change of the target rank can induce a significant increase of the reconstruction error due to the nonconvex nature of them''.
>
> >There is a typo or a problem of visualization in Figure 1. The KL distance should be between P and $\overline{P}$.
>
> We appreciate your careful reading. We think this may be caused by your pdf browser problem. In Figure 1, the KL distance from P to $\overline{P}$ is already written in the initial version.
>
> >In section 2.5, I think the following sentence needs to be fixed: “any tensor P can be represented by ...” → “any tensor P can be approximated by ... ”, because in general not any tensor can exactly represented as a rank-1 tensor.
>
> We keep this sentence because this is correct; if (and only if) all n($\geq$2)-body parameters are zeros, the rank of the corresponding tensor is 1.

---

> > ### Comment · Reviewer_w947 · 2022-12-09
> > **feedback provided**
> >
> > Dear authors,
> > Thank you for providing responses and clarifications to each of my criticisms and thanks for considering to change the title.
> > I am satisfied with all responses except with the one regarding the experimental section which I think is the weakest aspect of the paper.
> > However, I keep my score since I found the paper novel, useful and well written.

---

> > > ### Author Response · Authors · 2022-12-11
> > > **Authors' response to Reviewer w947**
> > >
> > > Thank you for your response.
> > > We appreciate your acknowledging our contributions.

---

### Official Review · Reviewer_GSHo · 2022-10-24

**Confidence:** 3
**Correctness:** 3
**Technical Novelty And Significance:** 4
**Empirical Novelty And Significance:** 3
**Recommendation:** 8

**Clarity, Quality, Novelty And Reproducibility:**

**--- Clarity/Quality ---**

The quality of writing is overall good, but clarity is lacking. This is in fact the main issue of the paper; see details under weaknesses.

A few other more minor points relating to clarity/quality:
- The 3rd listed contribution in Sec 1 ("We empirically show that ... reconstruction errors.") is vague. What does "more efficient" mean? Fewer parameters, less computation time?
- In Eq (7), the examples are, respectively, 1-, 2- and 3-body parameters, right? It would be good if this was clarified.
- Above Eq (10), by "substituting 0 for energies greater than 2nd and 3rd order", do you mean setting H^{(l_1, \ldots, l_n)}_{i_{l_1}, \ldots, i_{i_n}} = 0 for  $n \geq 2$ and $n \geq 3$, respectively? Please clarify.
- Below Eqs (10), (11), should the terms $/6$ and $/4$ in the definition of $Z$ be removed for correct scaling, since you're already taking 6th and 4th order roots?
- On page 5, the statement "$\mathcal{P}$ is approximated by the elementwise product of an $n$-order tensor for $n=2$ and $n=3$" is confusing. This makes it sound like you're doing Hadamard product between 2- and 3- way tensors, which would result in a 2- or 3-way tensor $\mathcal{P}$.
- Below Eq (15), should $|\mathcal{B}|/|\mathcal{R}|$ be $I/R^2$?
- Below Eq (18): reconstracted -> reconstructed

**--- Novelty ---**

The proposed framework seems highly novel to me. This is the main strength of the paper.

**--- Reproducibility ---**

It is hard to fully understand the method from the paper. There is code, but it doesn't appear to be well-documented.

**Strength And Weaknesses:**

**--- Strengths ---**

**S1.** The novelty of the proposal is a strong strength. I haven't seen anything like this proposal before. I think Sec 2.5 explains the novelty in a nice way: The framework does rank-1 decompositions, where the notion of "rank-1" is now generalized beyond the traditional tensor rank.

**S2.** The fact that the decomposition corresponds to a *convex* problem is another considerable strength, since most other tensor decompositions correspond to notoriously difficult non-convex optimization problems.

**S3.** The fact that no rank parameters need to be chosen is another important strength. Choosing an appropriate rank is typically also difficult for more traditional tensor approaches. Intuitively, it seems like it might be easier to choose appropriate levels of interaction between modes in the proposed framework than it is to choose rank in more traditional approaches.

**S4.** Although the experiments are not very comprehensive, the speed-up results in the few experiments that are provided are impressive.

**--- Weaknesses ---**

The proposal in the paper is very interesting. However, unfortunately there are points that aren't explained properly and that are difficult to follow.

**W1.** The optimization procedure---arguably a key advantage of the method---is difficult to understand.
- Concepts like "natural parameters", "flats" and "natural gradient" are never properly defined. Since most of the ICLR audience are unlikely to be familiar with these concepts, they should be explained. If there's not enough room in the main paper, this could be put in the appendix.
- The vectors $\theta^B$ and $\eta^B$ are, as far as I can tell, length-$|\mathcal{B}|$ vectors that contain the entries from $\theta$ and $\eta$ corresponding to indices in $B$, i.e., indices for which $\theta$ is zero: $\theta_b = 0$ for all $b \in B$. With this in mind, it seems like the stepping scheme in Eq (6) only updates entries in $\theta$ that are already zero. For a distribution $Q_1$ to lie in the e-flat as shown in Fig 1 (a), doesn't all entries $\theta_b$ for $b \in B$ have to be fixed to zero? In that case, why are those entries updated in Eq (6)? It seems like all this optimization procedure does is update parameters we know should be zero, while leaving the other parameters unchanged...
- The complexity analysis in Sec 2.6 is hard to follow since it doesn't refer back to the particular computational steps discussed earlier.
- What termination criteria is used in the gradient descent method? This is important when interpreting the experiment results.

**W2.** The practical connection between the $\theta$ parameters (e.g., in Eq (7)) and the $H$ parameters (e.g., Eq (9)) isn't explained well. The optimization is done over the parameters in $\theta$, but the conditions (e.g., set $n$-body interactions to zero for $n \geq 3$) are imposed on the $H$ parameters, which are sums of $\theta$ parameters. It is not clear to me how such conditions on the $H$ parameters translate to conditions on the $\theta$ parameters. For example, if $H_{i_k, i_m}^{(k,m)}$ is zero, then what indices should be added to the zero-index set $B$? The parameter count in Eq (15) counts the number of parameters in the $H$ factors---how does this relate to the number of $\theta$ parameters that are actually being optimized?

**W3.** The comparison between specific $n$-body approximations and conventional tensor decompositions is vague. For example, in Sec 3 you say "if we impose that decomposed factors can be represented as products with hyper-diagonal tensors $\Omega$, this decomposition is equivalent to a cyclic two-body approximation." It's not clear to me what this means. How should the symbol $\simeq$ in the figures be interpreted? Since fitting a tensor ring is a non-convex problem, it's clear that they can't be equivalent.

**W4.** Examples of things throughout the text that aren't well-defined/hard to follow:
- Sec 2.1: "dual flatness and orthogonality" of coordinate systems. What does this mean?
- Sec 2.3: "element-wise cyclic product of matrices".
- While terms like "partition function" may be standard in physics, they will be unfamiliar to most members of the ICLR audience.

**Summary Of The Paper:**

The paper proposes a new kind of tensor decomposition (TD) framework for non-negative tensors which is inspired by many-body interactions in physics. Unlike most traditional TDs which are hard to fit since they correspond to non-convex optimization problems, the proposed framework corresponds to a *convex* optimization problem. The authors propose a gradient descent-based scheme for solving it.
Another notable feature of the proposed framework is that it *does not* have a notion of rank---a parameter which is difficult to choose in more standard TDs.

### Update after rebuttal ###

The authors have done a good job of addressing all the issues I brought up in my original review. The optimization scheme makes sense now. What the various n-body constraints imply in terms on constraints on the $\theta$ parameters has also been clarified. The connections to other decompositions is clearer. The explanation of physics terms has also been improved which makes the paper more readable for non-physicists.

With these improvements, I think the paper is good enough for publishing in ICLR. I have increased my score from 3 to 8.

A few minor things I noticed when reading through the paper again after the rebuttal that the authors can fix in the camera ready version as appropriate:
- When mentioning natural parameters and dual flatness for the first time, it would be great if you provided a reference to the supplement where these things are discussed more in depth.
- In Eq (8), the order of the two summations should be changed, right? The first one should be $\sum_{k=1}^D \sum_{m=1}^{k-1}$ instead of what it is now, and the second one should be $\sum_{k=1}^D \sum_{m=1}^{k-1} \sum_{p=1}^{m-1}$.
- In sentence above Eq (10), there's a word missing (indicated in bold): "If all energies greater than 2nd-order or those *greater* than 3rd-order...".
- In the various places where you refer to the supplementary material, it would be great if you indicated which section in the supplementary material.
- Section "Synthetic data" at the bottom of page 8, there's a stray period that should be removed. ("...are shown in Figure 4 (a)*.*, and those...")
- Section "Real data" on page 9: You should refer to Fig 5 at some point, since it contains the results for this experiment.
- On page 14, in one of the paragraph headings: "Flattness" -> "Flatness"

**Summary Of The Review:**

I think the proposal itself is novel and interesting. It has many benefits like a convex optimization formulation and a lack of rank parameters to tune. Additionally, the experiment results (speed-ups in particular) are promising. These considerable strengths are unfortunately brought down by lack of clarity in the text. For example, the gradient descent optimization scheme is hard to follow, the connection between various parameters is not explained properly, and the connection between the presented framework and more traditional decompositions is not explained well. Also, the use of various physics terms that are non-standard in ML and numerical linear algebra add to the confusion.

I think once these clarity issues are sorted out, this will be a strong and interesting paper. I encourage the authors to work on improving the clarity, and I hope that my comments will be helpful for doing that.

---

> ### Author Response · Authors · 2022-11-19
> **Authors' response to Reviewer GSHo 1/**
>
> We appreciate many productive comments.
>
> **Strength And Weaknesses(1/):**
>
> >**W1.** Concepts like "natural parameters", "flats" and "natural gradient" are never properly defined. Since most of the ICLR audience are unlikely to be familiar with these concepts, they should be explained. If there's not enough room in the main paper, this could be put in the appendix.
>
> We have added more detailed explanation of these concepts, mainly used in information geometry, in Appendix D. In addition, since natural gradient is just a gradient method with second-order derivative in non-Euclidian space, we have modified the description in Section 2.1 as follows:
>
> Using natural gradient → Using gradient descent with second-order derivative
>
> >**W1.** W1. The vectors $\theta^B$ and $\eta^B$ are, as far as I can tell, length-|B| vectors that contain the entries from $\theta$ and $\eta$ corresponding to indices in $B$, i.e., indices for which $\theta$ is zero: $\theta_b=0 $ for all $b\in B$. With this in mind, it seems like the stepping scheme in Eq (6) only updates entries in $\theta$ that are already zero. For a distribution $Q_1$ to lie in the e-flat as shown in Fig 1 (a), doesn't all entries $\theta_b$ for $b\in B$ have to be fixed to zero? In that case, why are those entries updated in Eq (6)? It seems like all this optimization procedure does is update parameters we know should be zero, while leaving the other parameters unchanged…
>
> We appreciate your careful reading. We apologize that there were typos and they cause confusion. In section 2.1.1., we define $B$ as a set of indices for which $\theta$ parameters are imposed to be 0. However, it should be a set of indices for which $\theta$ parameters are **NOT** imposed to be 0. For a distribution $Q_1\in \mathcal{B}$, entries $\theta_b$ for $b\notin B$ have to be fixed to zero and entries $\theta_b$ for $b\in B$ are not be fixed to zero and updated in Eq.(6). The first paragraph in Section 2.1.1 are fixed as follows:
>
> ``Let $B$ be the set of indices of $\theta$ parameters that are not imposed to be $0$. Then Legendre decomposition coincides with a projection of a given nonnegative tensor $\mathcal{P}$ onto the subspace $\mathcal{B}=\set{ \theta\mid\theta_{i_1,\dots,i_D}=0 \text{ if } (i_1,\dots,i_D) \notin B\}.$''
>
> >**W1.** The complexity analysis in Sec 2.6 is hard to follow since it doesn't refer back to the particular computational steps discussed earlier
>
> In many-body approximation, in each iteration, the expectation parameter $\eta$ is updated by Equation (3), the Fisher-information matrix $G$ by Equation (5), the natural parameter $\theta$ by Equation (6), and the tensor $\mathcal{P}$ by Equation (3). Among these updates, the bottleneck is the update of $\theta$, which requires the computation of the inverse matrix of $G$. This is why we discuss only the computational complexity of the matrix inversion. We have added the following sentence in Sec 2.6.
>
> ``In many-body approximation, the overall complexity is dominated by the update of $\theta$, which includes matrix inversion of $G$.''
>
> >**W1.** What termination criteria is used in the gradient descent method? This is important when interpreting the experiment results
>
> It is simply terminated if the update is small enough. For the actual value, we follow the original implementation given by the authors of Legendre Decomposition [1], which is $||\boldsymbol\eta_t^B-\boldsymbol{\hat\eta}^B || < 10^{-5}$, where $\boldsymbol\eta^B_t$ is the expectation parameters on $t$-th step and $\boldsymbol{\hat\eta}^B$ is the expectation parameters of an input tensor, which are defined in Section 2.1.1.
>
> We have added the above terminate condition in Appendix B.
> - [1] Sugiyama, M., Nakahara, H., \& Tsuda, K. (2018). Legendre decomposition for tensors. Advances in Neural Information Processing Systems, 31.

---

> > ### Author Response · Authors · 2022-11-19
> > **Authors' response to Reviewer GSHo 2/**
> >
> > **Strength And Weaknesses(2/2):**
> >
> > >**W2.** The practical connection between the $\theta$ parameters (e.g., in Eq (7)) and the H parameters (e.g., Eq (9)) isn't explained well. The optimization is done over the parameters in $\theta$, but the conditions (e.g., set n-body interactions to zero for $n\geq3$) are imposed on the H parameters, which are sums of $\theta$ parameters. It is not clear to me how such conditions on the H parameters translate to conditions on the $\theta$ parameters. For example, if $H^(k,m)_{ik,im}$ is zero, then what indices should be added to the zero-index set B? The parameter count in Eq (15) counts the number of parameters in the H factors---how does this relate to the number of $\theta$ parameters that are actually being optimized?
> >
> > We have introduced $H$ to simply denote the total amount of each $n$-body interaction. Here we provide an example for ignoring two-body interaction between mode-k and mode-m. In this case, we need to set $H_{i_k,i_m}^{(k,m)} = 0$ for all $i_k \in \set\{1,\dots,I_k\}$ and $i_m \in \set\{1,\dots,I_m\}$. To do that, we set all $\theta_{i_k,i_m}^{(k,m)}=0$ for all $i_k \in \set\{2,\dots,I_k\}$ and $i_m \in\set \{2,\dots,I_m\}$.
> >
> > We added the description at the end of Section 2.2 as follows:
> >
> > ``In the following section, we reduce some of $n$-body interactions, that is, $H_{i_{l_1},\dots,i_{l_n}}^{(l_1,\dots,l_n)} = 0$, by fixing each parameter $\theta_{i_{l_1},\dots,i_{l_n}}^{(l_1,\dots,l_n)}=0$ for all indices $(i_{l_1},\dots,i_{l_n})\in\set\{2,\dots,I_{l_1}\}\times\dots\times\set\{2,\dots,I_{l_n}\}$.''
> >
> > >**W3.** The comparison between specific n-body approximations and conventional tensor decompositions is vague. For example, in Sec 3 you say "if we impose that decomposed factors can be represented as products with hyper-diagonal tensors $\Omega$, this decomposition is equivalent to a cyclic two-body approximation." It's not clear to me what this means. How should the symbol $\simeq$ in the figures be interpreted? Since fitting a tensor ring is a non-convex problem, it's clear that they can't be equivalent.
> >
> > You are right that tensor ring decomposition is a nonconvex optimization problem while cyclic two-body approximation is a convex optimization problem. These cost functions to be optimized are different. However, what we are saying is that the cyclic two-body approximation can be viewed as a tensor ring decomposition with constraints. Tensor ring decomposition decomposes a given tensor into some core tensors $\chi^{(d)}$ for $d=\{1,2\dots,D\}$. If we assume each element in the core tensor can be written as $\chi_{i,j,k}^{(d)} = \sum_m X_{i,m}\Omega_{m,j,k}$, these cost function becomes equivalent. We have added a more detailed description of the correspondence between these models in Appendix A. The symbol $\simeq$ means approximation. For example, Figure 2(b) describes that tensor ring decomposition approximates 4th order tensor $\mathcal{P}$ by products of four tensors.
> >
> > >**W4.** Sec 2.1: "dual flatness and orthogonality" of coordinate systems. What does this mean?
> >
> > Both of them are general properties of $\theta$- and $\eta$- coordinate systems in information geometry. The entire space of probability distributions is e-flat and m-flat simultaneously, which is called dual flatness. The relation of fisher information matrix and its inverse matrix, $G_{u,v}=\partial \eta_u/\partial \theta_v$ and $(G^{-1})_{u,v}=\partial \theta_u/\partial \eta_v$, is called orthogonality. To avoid confusion, we have added these descriptions in Appendix D.
> >
> > >**W4.** While terms like "partition function" may be standard in physics, they will be unfamiliar to most members of the ICLR audience.
> >
> > Yes, the partition function is standard in physics, while it is also often used in the machine learning community [2, 3, 4]. We replaced ''partition function'' by ''partition function, or the normalization factor'' in Section 2.3. We believe it is also worthwhile to keep the relation to statistical physics where many-body approximation has been developed.
> >
> > - [2] Desjardins, G., Bengio, Y., \& Courville, A. C. (2011). On tracking the partition function. Advances in neural information processing systems, 24.
> > - [3] Murphy, K. P. (2012). Machine learning: a probabilistic perspective. MIT press.
> > - [4] Goodfellow, I., Bengio, Y., \& Courville, A. (2016). Deep learning. MIT press.
> >
> > **Clarity/Quality(1/2):**
> >
> > >The 3rd listed contribution in Sec 1 ("We empirically show that ... reconstruction errors.") is vague. What does "more efficient" mean? Fewer parameters, less computation time?
> >
> > It means faster computation with competitive reconstruction errors. We have modified as follows:
> >
> > ``We empirically show that many-body approximation is faster than low-rank approximation with competitive reconstruction errors.''

---

> > > ### Author Response · Authors · 2022-11-19
> > > **Authors' response to Reviewer GSHo 3/3**
> > >
> > >
> > > **Clarity/Quality(2/2):**
> > >
> > > >In Eq (7), the examples are, respectively, 1-, 2- and 3-body parameters, right? It would be good if this was clarified.
> > >
> > > Yes, they are. We have added the following senescence below Eq.~(7):
> > >
> > > ``for $n=1,2$, and $3$, respectively.''
> > >
> > > >Above Eq (10), by "substituting 0 for energies greater than 2nd and 3rd order", do you mean setting $H_{l_1, \ldots, i_{i_n}}^{(l_1, \ldots, l_n)} = 0$ for $n>2$  and $n>3$, respectively? Please clarify.
> > >
> > > Your understanding is correct. To avoid confusion, we have fixed the sentence as follows:
> > >
> > > ``If all energies greater than 2nd-order or those than 3rd-order in equation (8) are ignored, that is, $H_{i_{l_1},\dots,i_{l_n}}^{(l_1,\dots,l_n)} = 0$ for $n>2$ or $n>3$, $\mathcal{P}$ is approximated as follows...''
> > >
> > > >Below Eqs (10), (11), should the terms /6 and /4 in the definition of be removed for correct scaling, since you're already taking 6th and 4th order roots?
> > >
> > > Thank you for your careful reading. You are right. We have also fixed it below Eq.(13).
> > >
> > > >On page 5, the statement "P is approximated by the elementwise product of an n-order tensor for n=2 and n=3" is confusing. This makes it sound like you're doing Hadamard product between 2- and 3- way tensors, which would result in a 2- or 3-way tensor P.
> > >
> > > This is not Hadamard product and we agree this statement is causing confusion. What we wanted to emphasize is that the sum symbol $\sum$ does not appear in the notation after the approximation. This is a different property from traditional low-rank approximations. We have removed the sentence and added the following description in the first paragraph of Section 2.3:
> > >
> > > ``As we see below, approximated tensors are represented without the summation symbol $\sum$. This property is different from existing low-rank approximations except for rank-1 approximation.''
> > >
> > > >Below Eq (15), $|B|/|R|$ should be $I/R^2$ ?
> > >
> > > You are correct. It should have been $I/R^2$ instead of $DI/R^2$. We have fixed it.
> > >
> > > >Below Eq (18): reconstracted → reconstructed
> > >
> > > We have fixed it.
> > >
> > > **Reproducibility :**
> > >
> > > >It is hard to fully understand the method from the paper. There is code, but it doesn't appear to be well-documented.
> > >
> > > As we described above, we had a typo in Section 2.1.1 and we guess it was the main reason for the confusion. We have corrected these typos and provided the pseudo-code in Appendix B.
> > >
> > > **Summary of the Review  :**
> > >
> > > >The connection between various parameters is not explained properly, and the connection between the presented framework and more traditional decompositions is not explained well.
> > >
> > > We have added an explanation that we enforce all n-body parameters to be zero to remove n-body energy at the end of Section 2.1. We have added Appendix A to describe the relationship between cyclic two-body approximation and tensor ring decomposition.
> > >
> > > We hope this improves the clarity of the paper and helps understanding.

---

> > > > ### Comment · Reviewer_GSHo · 2022-11-19
> > > > **Thank you for fixing issues --- I have increased my score**
> > > >
> > > > Thank you for fixing the issues I brought up. I have increased my score from 3 to 8.
> > > >
> > > > There are a few additional minor issues I noticed when reading through the paper again. I hope you can address these for the camera ready version:
> > > > - When mentioning natural parameters and dual flatness for the first time, it would be great if you provided a reference to the supplement where these things are discussed more in depth.
> > > > - In Eq (8), the order of the two summations should be changed, right? The first one should be $\sum_{k=1}^D \sum_{m=1}^{k-1}$ instead of what it is now, and the second one should be $\sum_{k=1}^D \sum_{m=1}^{k-1} \sum_{p=1}^{m-1}$.
> > > > - In sentence above Eq (10), there's a word missing (indicated in bold): "If all energies greater than 2nd-order or those *greater* than 3rd-order...".
> > > > - In the various places where you refer to the supplementary material, it would be great if you indicated which section in the supplementary material.
> > > > - Section "Synthetic data" at the bottom of page 8, there's a stray period that should be removed. ("...are shown in Figure 4 (a)*.*, and those...")
> > > > - Section "Real data" on page 9: You should refer to Fig 5 at some point, since it contains the results for this experiment.
> > > > - On page 14, in one of the paragraph headings: "Flattness" -> "Flatness"

---

> > > > > ### Author Response · Authors · 2022-11-21
> > > > > **Authors' response to Reviewer GSHo**
> > > > >
> > > > > We appreciate your careful reading of our response and paper again, and thank you for increasing the score. We promise to correct all the additional typos you pointed out in the camera-ready version. Your review is quite helpful in improving the overall quality of the paper.

---

### Official Review · Reviewer_QF9P · 2022-10-27

**Confidence:** 4
**Correctness:** 2
**Technical Novelty And Significance:** 2
**Empirical Novelty And Significance:** 2
**Recommendation:** 3

**Clarity, Quality, Novelty And Reproducibility:**

This paper performs a Legendre tensor approximation as a many-block tensor approximation.  It also introduces a tensor network interaction diagrams for visualization.

The authors' approach has many similarities to causal block multilinear decomposition.  A block tensor decomposition splits a data tensor into $N$ tensor blocks that have their own individual intrinsic mode matrices and share a subset of their extrinsic mode matrices.  The data blocks "interact" with each other to compute the mode matrix representations for the extrinsic modes that they share.   The zeros in a compound/hierarchical core tensor make the interactions self-evident.

See: http://web.cs.ucla.edu/~maov/CausalX_Block_Multilinear_Factor_Analysis_ICPR2020.pdf

See Fig 4b: http://web.cs.ucla.edu/~maov/CausalDeepLearning.pdf


The authors employ abstract notation which would benefit from being grounded in an application.  Without such grounding, it isn't always clear what the variables represent.

According to Appendix B, the authors seem to map the Dino light field data, for example, into a "many-body" approximation problem by reshaping a 5th-order data tensor (9 x 9 x 512 x 512 x 3)  into a 9th-order tensor (6 x8 x 6 x 8 x 6 x 8 x 6 x 8 x 12) and splitting the new tensor into 7 different 9th order tensors that are decomposed by 7 tensor rings with different ring ranks.  Reporting this information is nice, but one still needs an explanation.   Why is the 7-block subdivision of the Dino LFD a "many-body" problem?

Does it matter what type of reshaping or subdivision is performed?  Would any arbitrary reshaping and subdivision of a data tensor work equally well?  Which modes were considered dominant versus irrelevant in the 4D light field example?  Which modes were dimensionally reduced?  Are the reconstructed images still usable?

The network diagrams use both square and circle nodes that are labeled with indices, matrices, and tensors. Some are filled in and others are not.  Not entirely clear on the difference.


Detailed comments:
-----
The introduction and the statements that motivate the work need some editing:

+ "Tensor decomposition is one of the most popular methods that extract features by approximating tensors by the sum of products of smaller size of tensors, often called factors."

Why are the terms of a summation called factors? "Factorization or factoring consists of writing a number or another mathematical object as a product of several factors, usually smaller or simpler objects of the same kind."   A tensor decomposition, including the CP decomposition, consists of writing a data tensor as the product of a core tensor and a set of mode matrices. Mode matrices are called causal factors when performing causal inference.

+ "In most of tensor decomposition approaches, a low-rank structure is typically assumed, where a given tensor is approximated by a linear combination of a small number of bases. Such decomposition requires the following two information. First, it requires the structure, which specifies the type of decomposition such as CP decomposition (Hitchcock, 1927) and Tucker decomposition..."

As written, the authors seem to categorize both the CP and the Tucker decompositions as rank decompositions. The Tucker and Candecomp/Parafac (CP) tensor decomposition embody different properties of the matrix SVD.  The CP decomposition computes the best fitting K rank-1 terms for a user specified $K$.  The Tucker decomposition computes a set of orthonormal mode matrices and may be thought of as performing a multilinear rank -- rank-$(R_1,\dots, R_M)$.  One can write a CP decomposition using a product of terms similar to Tucker. Tucker can be written as a sum of rank-1 terms. However, a CP is not an orthonormal mode matrix computation and Tucker is not a rank decomposition.  FYI, CP is not a rank decomposition either.

+ "In recent years, tensor networks (Cichocki et al., 2016) have been introduced, which can intuitively and flexibly design the structure including tensor train decomposition (Oseledets, 2011)..."

Hmmm ...  Tensor networks are a particular type of visualization of tensor algebra data analysis.  Tensor Ring is a sequential version of the M-mode SVD which performs a parallel Tucker.  Tensor Train is a sequential one-time iteration of Tensor Ring.

See sec 5.2: https://arxiv.org/abs/1606.05535
See fig. 3f: http://web.cs.ucla.edu/~maov/CausalDeepLearning.pdf

+ How is an m-projection computed? What is an e-flat?
+ What do the vertical red dashed lines in Fig 4 and 5 mean?
+ Why use natural gradients? Why not use simulated annealing or stochastic gradient descent?
+ What are natural parameters?
+  However, the indices $i_d$, $i^{'}_d$, $j_d$ and $I_d$  are used inconsistently.  In tensor algebra, upper case italics are used to represent upper bounds. For example, $1\le d\le D$ and $1\le r_d \le {\bar R}_d \le R_d$.
In eq 1. $i^{'}_d$
 is used as a variable and $i_d$ is used as an upper bound, but in the text above the upper bound is $I_d$ . The Mobius function uses $j_d$ as a variable, $i_d$ as a lower bound and $i^{'}_d$
 as an upper bound.

+ Please define $\mu$  What do the superscript and subscripts mean?

+ "a tensor as an undirected graph, whose nodes correspond to matrices or tensors and edges are the modes of summation in tensor products"

I see both square and circle nodes that are labeled with indices, matrices and tensors. Some are filled in and others are not. What is the difference?  I would like to see every edge and node labeled with the mathematical operators and variable names.

Missing References :

+ Shashua and Hazan wrote the first non-negative tensor decomposition paper.

@inproceedings{shashua2005non,
  title={Non-negative tensor factorization with applications to statistics and computer vision},
  author={Shashua, Amnon and Hazan, Tamir},
  booktitle={Proceedings of the 22nd international conference on Machine learning},
  pages={792--799},
  year={2005}
}

+ Vasilescu and Terzopoulos introduced tensor factor analysis for computer vision, computer graphics and machine learning and decomposed data tensors into their causal factors of data formation.  The TensorFaces paper introduces a parallel Tucker decomposition -- the M-mode SVD (MPCA).

@inproceedings{Vasilescu2002,
  author =	 "M. A. O. Vasilescu and D. Terzopoulos",
  fullauthor =	 "M. Alex O. Vasilescu and Demetri Terzopoulos",
  title =	 "Multilinear Analysis  of Image Ensembles: {T}ensor{F}aces",
  booktitle =	 "Proc. European Conf. on Computer Vision (ECCV 2002)",
  address =	 "Copenhagen, Denmark",
  month =	 "May",
  year =  "2002",
  pages = "447-460"
}

+ Multilinear projection estimates the causal factors of data formation from unlabeled test image(s) given an estimated forward model:

@INPROCEEDINGS{Vasilescu2011,
author={M. A. O. Vasilescu},
booktitle={Proc. IEEE Inter. Conf. on Automatic Face Gesture Recognition (FG 2011)},
title={Multilinear projection for face recognition via canonical decomposition},
year={2011},
pages={476-483},
doi={10.1109/FG.2011.5771445},
month={Mar},}

+ Tensor causal factor analysis paper that split a data tensor into data blocks and specifies which data blocks share the same mode matrix and hence contribute to the mode matrix computation.

@inproceedings{Vasilescu20,
    author={Vasilescu, M. Alex O. and Kim, Eric and Zeng, Xiao S.},
    booktitle={2020 25th International Conference of Pattern Recognition (ICPR 2020)},
    title={Causal{X}: Causal e{X}planations and Block Multilinear Factor Analysis},
    year={2021},
    location={Milan, Italy},
    month={Jan},
    pages={10736--10743},
    url={http://web.cs.ucla.edu/~maov/CausalX_Block_Multilinear_Factor_Analysis_ICPR2020.pdf}

+ Panagakis etal 2021 are describing papers that are performing tensor regression. The paper contains multiple misconceptions that have been floating around in the ML tensor community.  The authors may wish to address those misconceptions.   Alternatively, the authors may want to reference other cleaner papers by the same authors, such as:

 @article{Kossaifi20,
  author  = {Jean Kossaifi and Zachary C. Lipton and Arinbjorn Kolbeinsson and Aran Khanna and Tommaso Furlanello and Anima Anandkumar},
  title   = {Tensor Regression Networks},
  journal = {Journal of Machine Learning Research},
  year    = {2020},
  volume  = {21},
  number  = {123},
  pages   = {1-21},
}

+ Original light field/lumigraph papers:

@inproceedings{gortler1996lumigraph,
  title={The lumigraph},
  author={Gortler, Steven J and Grzeszczuk, Radek and Szeliski, Richard and Cohen, Michael F},
  booktitle={Proceedings of the 23rd annual conference on Computer graphics and interactive techniques},
  pages={43--54},
  year={1996}
}

@inproceedings{levoy1996light,
  title={Light field rendering},
  author={Levoy, Marc and Hanrahan, Pat},
  booktitle={Proceedings of the 23rd annual conference on Computer graphics and interactive techniques},
  pages={31--42},
  year={1996}
}

TensorTexture models the bidirectional texture function (a type of 6D light field - images of a scene are captured as the viewpoint and illumination were varied).  This is the first tensor paper in computer graphics.

@article{Vasilescu04,
  author =	 "M. A. O. Vasilescu and D. Terzopoulos",
  title =	 "{T}ensor{T}extures: {M}ultilinear Image-Based Rendering",
  journal =	 "ACM Transactions on Graphics",
  volume =	 "23",
  number =	 "3",
  month =	 "Aug",
  year =	 "2004",
  pages=	 "336-342",
  note =     "Proc. ACM SIGGRAPH 2004 Conf., Los Angeles, CA",
}

**Strength And Weaknesses:**

Strengths:
1. This modified non-negative tensor approximation subdivides a data tensor into multiple tensor blocks that approximate the tensor whole.  The tensor blocks have modes that are entirely their own, and modes that they share with other tensor blocks.

2. Mode interaction is visualized through a new network diagram.

Weakness:
1. There is no algorithmic description.  From Appendix B, I see that a tensor is reshaped and split into several tensor blocks each approximated by a tensor ring. Presumably, some of the irrelevant mode interactions between the tensor rings (tensor blocks) are set to zero.  Which mode interactions were set to zero? Would any reshaping and tensor block subdivision work equally well?

2. The authors have not clearly articulated

+ (i) the criteria for splitting a data tensor into $N$ data blocks,
+ (ii) the criteria for determining which mode interactions are deemed to be dominant or irrelevant and set to zero, or
+ (iii) the choice of tensor ring ranks.

4. It is unclear what the experimental results are demonstrating.  What is the AI, ML, CV, or CG problem being solved?   The goal is to find meaningful representations that increase recognition rates or result in high-quality reconstructed or synthesized images, etc.  Reconstruction errors are not always good proxy indicators for the previously mentioned tasks. For CG, the authors may wish to display reconstructed or newly synthesized images versus reconstruction errors versus decomposition speed versus reconstruction speed.

If the goal is compression, this should be stated.  The authors may wish to compare their work with jpeg or other state-of-the-art compression techniques.

**Summary Of The Paper:**

The authors' starting point is the "Legendre Decomposition for Tensors" paper. The Legendre decomposition is modified to approximate a data tensor by "dividing" the original data tensor N-ways and approximating the N-blocks which "interact" with each other through a subset of the tensor modes. The authors assume that tensor blocks have dominant interactions while others can be deemed to be irrelevant and set to zero. The authors refer to this modified approximation as a many-body approximation.  The approach was tested on simulation data and 4D light field data etc. The authors report on some of the experiments lower reconstruction errors relative to prior research.




**Summary Of The Review:**

The paper introduces a modified Legendre decomposition that subdivides a data tensor into multiple tensor blocks to approximate a tensor whole.  It also introduces an interaction tensor network diagram for better visualization.  However, the paper is missing an algorithm, a concrete example, some important discussions, and comparisons to prior work.

I look forward to reading the final version of this paper when all outstanding issues discussed above are fixed.

______
______
______
______
______
---  UPDATED REVIEW  given the authors' response:  ---
---
The paper's title, text, fig 3, and experiment description originally led me to believe that a tensor was approximated by multiple bodies  that interact through a subset of modes, such as the parts that make up a body. I assumed that the division into multiple bodies was achieved by setting some of the natural parameters of a modified Legendre decomposition to zero which was then followed by factorization of each body with a Legendre decomposition. The authors informed me that this was not the case.

The paper does state that it approximates a data tensor by formulating a "many-body approximation as a special case of Legendre decomposition by setting some of the natural parameters to be zero" (pg.2, paragraph 2).  However, the multi-body language is unrelated to the multi-bodies/particles one encounters in physics and unrelated to the multi-bodies/parts that make up an object.  Apparently, the "bodies" and "interactions" are artifacts of a sequential implementation that do not have a  physical meaning and do not hold true if one employs a parallel implementation.  Despite repeated requests, the authors have refused to define the meaning of a 2-body or N-body.  Why?

The name "interaction diagram" is a misnomer. A more appropriate name is an operation order diagram.   All the mode interactions are active in a Tucker decomposition which is self-evident in an M-mode SVD (parallel Tucker), but it is not immediately apparent if one looks at the operation order of a tensor ring (sequential Tucker). In Fig. 2, "the mode interaction diagram" for a Tucker decomposition depicts the operation order of a sequential implementation, ie a tensor ring.

The algorithm in the supplement is the natural gradient descent algorithm that does not integrate equations 1,3,5.  The equations are in the main body of the paper and contain undefined variables that would have had to be specified if they were part of the algorithm.  It would have been nice if the authors would have explained the meaning of the superscripts and subscripts which can vary from author to author.

The experiments are inconclusive. The authors have not solved any AI/ML/CV/CG problems. Hence, it is unclear if their approach is useful. The authors have asserted that their approximated images are still usable, but have not displayed any synthesized or reconstructed images.  Why?


Conclusion:
---
The authors' claim that they are performing a multi-body approximation is unsubstantiated.  The terms "multi-bodies" and "interactions"  are unrelated to the multi-bodies/particles in physics and these terms seem to be artifacts of a sequential optimization.  Beyond the inappropriate nomenclature that is misleading, the paper talks in generalities without being grounded in a concrete AI/ML/CV/CG example and without providing any intuition.  The algorithm is missing many details.

Currently, this paper is not ready for publication at ICLR.


Other comments:
---
1. The title is misleading and a type of flag planting.  The authors are not performing a multibody approximation where the bodies are physics particles or the parts of an object.  Hence, a more appropriate title might be "Legendre Tensor Decomposition envisioned as an M-body Approximation," and even this is a stretch.

2. > Our approach, describing interactions between modes using energy functions, is different from existing methods that focus on interactions between mode matrices (Vasilescu & Terzopoulos 2002; Vasilescu 2011) or block tensors (Vasilescu et al. 2007)

The above is inaccurate.  Object parts may not interact with each other at all, or may interact with each other through a subset of modes.  The following is a more accurate description:

Tensor factor analysis was introduced in computer vision with the TensorFaces papers (Vasilescu and Terzopoulos 2002; Vasilescu 2011) and in computer graphics with the TensorTexture paper (Vasilescu and Terzopoulos 2004). Vasilescu etal employ energy functions and focus on interactions between the many bodies that make up a tensor whole, such as the parts that make up an object (Vasilescu etal 2021; Vasilescu ICPR 2022).

By comparison, we define multi-body as  ...  (please provide a definition).

___
___

Given the other glowing review ratings, it is unlikely that there will be a reply forthcoming to any of my inquiries. Hence, I have gone ahead and just downgraded the paper. Best of luck and congratulations on your ICLR 2023 paper.

---

> ### Author Response · Authors · 2022-11-19
> **Authors' response to Reviewer QF9P 1/**
>
> We appreciate your valuable comments.
>
> **Strength And Weaknesses:**
>
> >There is no algorithmic description. From Appendix B, I see that a tensor is reshaped and split into several tensor blocks each approximated by a tensor ring.
>
> Although the algorithm of our method is already clear from Appendix A in our original submission, we have provided pseudo-code in Algorithm 1 in Appendix B according to your suggestion. Please note that our proposed method does not split a tensor into blocks. As described in Section 2.2 and Section 3, reshaping of tensors described in Appendix B is not for making tensor blocks but to reduce the computational complexity.
>
> >Presumably, some of the irrelevant mode interactions between the tensor rings (tensor blocks) are set to zero. Which ones? Would any reshaping and tensor block subdivision work equally well?
>
> As described above, our method does not divide given tensors into blocks.
>
> >The authors have not clearly articulated
> >- (i) the criteria for splitting a data tensor into data blocks
> >- (ii) the criteria for determining which mode interactions are dominant or irrelevant and set to zero, or
> >- (iii) the choice of tensor ring ranks.
>
> - (i) We do not split a data tensor into data blocks.
> - (ii) You are right that we do not discuss criteria for determining active interactions in the many-body approximation. This belongs to the model selection problem and is challenging, and it is out of the scope in our paper. In our experiments, we have focused on cyclic two-body approximation for our method as it can be viewed as an alternative to tensor ring decomposition. Since all one-body interactions and two-body interactions between neighbour modes are activated in cyclic two-body approximation, such criteria do not needed.
> - (iii) Our proposed method does not require tensor ring ranks, which is one of advantages of our method.
>
> >It is unclear what the experimental results are supposed to demonstrate. In CV, CG and ML the goal is to find meaningful representations that result in higher recognition rates or high-quality synthesized images. ML methods that employ either neural networks or tensor methods use overcomplete meaningful representations rather than low-rank methods.
>
> We share the same motivation with the existing low-rank approximation including tensor ring decomposition, whose usefulness has been already widely known in the ML community. Since our proposal is a constrained version of the existing low-rank approximation to make convex optimization possible, we aim at demonstrating the efficiency of our method in our experiments. Please note that our aim in this paper is not providing a new low-rank method with respect to a specific scenario like image compression in CV but proposing a new general and flexible formulation of low-rank approximation with theoretical support.
>
> >If the goal is compression, this should be stated. The authors may wish to compare their work with jpeg or other state-of-the-art compression techniques.
>
> Our goal is not compression, which is the same as existing tensor low-rank approximation methods.
>
> **Clarity, Quality, Novelty And Reproducibility(1/2):**
> >The authors' approach has many similarities to causal block multilinear decomposition. A block tensor decomposition splits a data tensor into N tensor blocks that have their own individual intrinsic mode matrices and share a set of extrinsic mode matrices. The data blocks "interact" with each other to compute the mode matrix representations for the extrinsic modes that they share. The zeros in a compound/hierarchical core tensor make the interactions self-evident.
> >
> >See : http://web.cs.ucla.edu/~maov/CausalX_Block_Multilinear_Factor_Analysis_ICPR2020.pdf
> >
> >See Fig 4b: http://web.cs.ucla.edu/~maov/CausalDeepLearning.pdf
>
> Although we acknowledge the information, since our proposed method is neither block decomposition nor hierarchical decomposition, it is not related to those papers. Those papers discuss interactions between block tensors, our method focuses on interactions between modes. To make it clear, we have added the following sentence in Introduction:
>
> ``Our approach, describing interactions between modes using energy functions, is different from existing methods that focus on interactions between mode matrices (Vasilescu \& Terzopoulos, 2002; Vasilescu, 2011) or block tensors (Vasilescu et al., 2021).''

---

> > ### Author Response · Authors · 2022-11-19
> > **Authors' response to Reviewer QF9P 2/**
> >
> > **Clarity, Quality, Novelty And Reproducibility(2/2):**
> > >According to Appendix B, the authors seem to map the Dino light field data, for example, into a "many-body" approximation problem by reshaping a 5th-order data tensor (9 x 9 x 512 x 512 x 3) into a 9th-order tensor (6 x8 x 6 x 8 x 6 x 8 x 6 x 8 x 12) and splitting the new tensor into 7 different 9th order tensors that are decomposed by 7 tensor rings with different ring ranks. Reporting this information is nice, but one still needs an explanation. Why is the 7-block subdivision of the Dino LFD a "many-body" problem? Does it matter what type of reshaping or subdivision is performed?
> >
> > As we have already discussed in Section 2.6 and stated “Each tensor has been reshaped to reduce the computational complexity.” in Section 3.2 in our initial submission, these reshaping for real datasets is just for reducing computational complexity and nothing to do with the many-body problem at all. Our method can be applied to such datasets even without reshaping. We do not subdivide data tensors. In Appendix C, the sentence (2, 3, 2, 2, 2, 2, 2, 2, 2), …, (2, 2, 2, 2, 2, 2, 2, 2, 9) describes the target rank for baseline methods.
> >
> > Although our method does not require them, we wrote these target ranks for reproducibility. To avoid confusion, we have added phrases ``For baseline methods,'' before these descriptions in the revised version.
> >
> > >Would any arbitrary reshaping and subdivision of a data tensor work equally well? Which modes were considered dominant versus irrelevant in the 4D light field example? Which modes were dimensionally reduced? Are the reconstructed images still usable?
> >
> > Again, our proposed method does not divide given tensors. If we reshape a tensor so that the number of parameters $|\mathcal{B}|$ becomes smaller, the computation will be faster, but the reconstruction error will be larger. If the reconstruction error is small, the reconstructed data is not significantly different from the original data. In such a case, the reconstructed data is usable after inverse reshaping.
> >
> > **Detailed comments(1/):**
> >
> > > ''Tensor decomposition is one of the most popular methods that extract features by approximating tensors by the sum of products of smaller size of tensors, often called factors''
> > > Why are the terms of a summation called factors? ``Factorization or factoring consists of writing a number or another mathematical object as a product of several factors, usually smaller or simpler objects of the same kind.''A tensor decomposition, including the CP decomposition, consists of writing a data tensor as the product of a core tensor and a set of mode matrices. Mode matrices are called causal factors when performing causal inference.
> >
> > Our sentence means that smaller tensors obtained by decomposition are called *factors* (not about the sum). This is a commonly used terminology in the tensor community, for example, we can find the following sentence:
> >
> > ``Tensor decomposition can break a large-size tensor into many small-size factors, which can reduce the storage and computation complexity during data processing."
> >
> > at the beginning of Chapter 2 in [1]. To avoid confusion, we rephrased the sentence as:
> > ``Tensor decomposition is one of the most popular methods that extract features by approximating tensors by the sum of products of smaller tensors. These smaller tensors are often called factors.''
> > Since we do not perform causal inference, the term "causality factor" is irrelevant.
> > - [1] Liu, Y., Liu, J., Long, Z., Zhu, C. (2022). Tensor Decomposition. In: Tensor Computation for Data Analysis. Springer, Cham.
> >
> > >"In recent years, tensor networks (Cichocki et al., 2016) have been introduced, which can intuitively and flexibly design the structure including tensor train decomposition (Oseledets, 2011)...”
> > >
> > >Hmmm ... Tensor Ring is a sequential version of the M-mode SVD which performs a parallel Tucker.  Tensor train is a sequential one-time iteration of Tensor Ring.
> > >
> > >See sec 5.2: https://arxiv.org/abs/1606.05535
> > >
> > >See fig. 3f: http://web.cs.ucla.edu/~maov/CausalDeepLearning.pdf
> > >
> > >Tensor networks are a particular type of visualization of tensor algebra data analysis, but it is not the only way.
> >
> > We understand it. Still, we believe that tensor networks are recognized as useful tools to intuitively design tensor decomposition, and we have claimed it in this sentence. The original paper on tensor ring decomposition that you provided also uses tensor networks to develop an intuitive argument.

---

> > > ### Author Response · Authors · 2022-11-19
> > > **Authors' response to Reviewer QF9P 3/**
> > >
> > >
> > > **Detailed comments(2/):**
> > > >"In most of tensor decomposition approaches, a low-rank structure is typically assumed, where a given tensor is approximated by a linear combination of a small number of bases. Such decomposition requires the following two information. First, it requires the structure, which specifies the type of decomposition such as CP decomposition (Hitchcock, 1927) and Tucker decomposition...”
> > > >
> > > >As written, the authors seem to categorize both the CP and the Tucker decompositions as rank decompositions. The Tucker and Candecomp/Parafac (CP) tensor decomposition embody different properties of the matrix SVD. The CP decomposition computes the best fitting K rank-1 terms for a user specified K. The Tucker decomposition computes a set of orthonormal mode matrices and may be thought of as performing a multilinear rank -- rank-(R1,…,RM). One can write a CP decomposition using a product of terms similar to Tucker. Tucker can be written as a sum of rank-1 terms. However, a CP is not an orthonormal mode matrix computation and Tucker is not a rank decomposition. FYI, CP is not a rank decomposition either.
> > >
> > > Thank you for your explanation. However, respectfully, we are confused as to the point that you raised for correction. We think it does not matter whether the representations are orthonormal or not in this context. What we want to say in this introduction is that many tensor decomposition approaches require a two-step process, first determining the low-rank structure, followed by determining the hyperparameter called rank.
> > >
> > > The parameter K appearing in your explanation is called the CP rank and $(R1,...,RM)$ is called the Tucker rank. Both CP and Tucker decomposition assume that a tensor can be expressed as a product of small tensors. We call structures that can be represented by a product of small tensors *low-rank structures*. The use of these technical terms is common in the tensor community. Here, we provide an example. We can find the following sentence *tensors often exhibit a low-rank structure, and can be approximated by a low-rank tensor factorization, such as CANDECOMP/PARAFAC(CP), tensor train, or Tucker factorization* in Introduction in [2].
> > >
> > > We kindly ask you to provide additional comments if we have missed something.
> > > - [2] Sun, Y., Guo, Y., Luo, C., Tropp, J., \& Udell, M. (2020). Low-rank tucker approximation of a tensor from streaming data. SIAM Journal on Mathematics of Data Science, 2(4), 1123-1150.
> > >
> > > >How is the m-projection computed? What is e-flat.
> > >
> > > The m-projection is minimization of Equation (4) by the natural gradient method. We have added pseudo-code of m-projection in Appendix B. e-flat is defined above Eq.(4) in our original submission.
> > >
> > > ``The subspace $\mathcal{B}$ is e-flat, meaning that the logarithmic combination, or called e-geodesic, $\mathcal{R} \in \{ (1-t) \log{\mathcal{Q}_1} + t\log{\mathcal{Q}_2} - \phi(t) \mid 0<t<1 \}$ of any two points $\mathcal{Q}_1,\mathcal{Q}_2 \in \mathcal{B}$ is included in the subspace $\mathcal{B}$, where $\phi(t)$ is a normalizer.''
> > >
> > > We agree that more explanation is helpful for the reader. Therefore we have added a section to describe these concepts in more details in Appendix D.
> > >
> > > >What does the vertical red dashed line in fig. 4 and 5 mean?
> > >
> > > The vertical red dashed line in Figures 4 and 5 denotes $|\mathcal{B}|$, the number of parameters used in cyclic two-body approximation. We have added the description in captions in Figures 4 and 5.
> > >
> > > >Why use natural gradients? Why not use simulated annealing or stochastic gradient descent?
> > >
> > > Thank you four your technical suggestion. As we can see in Figures 4 and 5, the natural gradient method is efficient, and it can always find the global optimal solution as our method is convex optimization. Therefore we do not need other optimization methods that you have listed.
> > >
> > > >What are natural parameters?
> > >
> > > We call $\theta$-parameters natural parameters, which is common terminology in information geometry. We have added detailed explanation in Appendix D.
> > >
> > > >In eq 1. $i'_d$ is used as a variable and $i_d$ is used as an upper bound, but in the text above the upper bound is $I_d$. The Mobius function uses $j_d$ as a variable, $i_d$ as a lower bound and $i_d'$
> > > as an upper bound.
> > >
> > > Our notation is consistent. To demonstrate that, we provide an example for $D=2$ and $I_1=I_2=2.$ The relation between tensor $\mathcal{P}$ and its parameter $\theta$ are given as follows:
> > >
> > > $P_{1,1}=\exp{} (\theta_{11})$, \
> > > $P_{1,2}=\exp{} (\theta_{11}+\theta_{12})$, \
> > > $P_{2,1}=\exp{} (\theta_{11}+\theta_{21})$, \
> > > $P_{2,2}=\exp{} (\theta_{11}+\theta_{21}+\theta_{12}+\theta_{22})$.
> > >
> > > To describe the relationship between $P$ and $\theta$, we have to write $P_{i,j} = \exp  \sum_{i'=1}^{i}  \sum_{j'=1}^{j} θ_{i',j'} $.

---

> > > > ### Author Response · Authors · 2022-11-19
> > > > **Authors' response to Reviewer QF9P 4/4**
> > > >
> > > > **Detailed comments(3/3):**
> > > >
> > > > >Please define $\mu$. What do the superscript and subscripts mean?
> > > >
> > > > The Möbius function $\mu$ is defined in the next equation to Eq.(2). The Möbius function is a mapping from $S \times S$ to $\set{-1,0,1}$, where $S$ is the index set, that is, $S = \set{1,\dots,I_1\} \times\dots \times\set{1,\dots,I_D\} $. Due to space limitations, we represent one of the two arguments by superscripts and the other by subscripts. We clarified the domain and the value range of the Möbius function in the main text to avoid confusion. We have also added examples in Appendix D
> > > >
> > > > >I see both square and circle nodes that are labeled with indices, matrices and tensors. Some are filled in and others are not. What is the difference?
> > > >
> > > > Each square with a label is a tensor. Each circle with a label is a mode. A black square is an interaction between modes, defined in Section 2.2. A black circle is a hyper diagonal tensor $\Omega_{ijk}=\delta_{ij}\delta_{jk}\delta_{ki}$, which is defined in Section 2.4. There is a typo that the tensor $\mathcal{P}$ in Figures 2(b) and 3(c) should have been enclosed in a square, but it was enclosed in a circle. We have fixed it in our revised version.
> > > >
> > > > >I would like to see every edge and node labeled with the mathematical operators and variable names.
> > > >
> > > > We have added variable names in Figures 2 and 3. In tensor networks, edges between tensors mean taking the sum with fixing the indices written in endpoints of edges. For example, for connected two matrices A and B, that is i-[A]-[B]-k, it means $\sum_{j}A_{ij}B_{jk}.$ In interaction representation, edges through $\blacksquare$ between modes mean existing interaction. For example, if there are connection, ($i_k$)-$\blacksquare$-($i_m$), it means $H^{(k,m)}_{i_k,i_m}\neq 0$.
> > > >
> > > > **Missing References:**
> > > >
> > > > Thank you for listing up a number of valuable references. We have carefully examined all the papers and picked up relevant papers from them and cited in our paper as stated in the following.
> > > >
> > > > >Shashua and Hazan wrote the first non-negative tensor decomposition paper.
> > > > >- @inproceedings{shashua2005non,
> > > >   title={Non-negative tensor factorization with applications to statistics and computer vision},
> > > >   author={Shashua, Amnon and Hazan, Tamir},
> > > >   booktitle={Proceedings of the 22nd international conference on Machine learning},
> > > >   pages={792--799},
> > > >   year={2005}}
> > > >
> > > > Thank you for your suggestion. We have cited it in the first paragraph of the Introduction.
> > > >
> > > > >Vasilescu and Terzopoulos introduced tensor factor analysis for computer vision, computer graphics and machine learning and decomposed data tensors into their causal factors of data formation. The TensorFaces paper introduces a parallel Tucker decomposition -- the M-mode SVD (MPCA).
> > > > > - @inproceedings{Vasilescu2002, author = "M. A. O. Vasilescu and D. Terzopoulos", fullauthor = "M. Alex O. Vasilescu and Demetri Terzopoulos", title = "Multilinear Analysis of Image Ensembles: {T}ensor{F}aces", booktitle = "Proc. European Conf. on Computer Vision (ECCV 2002)", address = "Copenhagen, Denmark", month = "May", year = "2002", pages = "447-460" }}
> > > > >
> > > > >Multilinear projection estimates the causal factors of data formation from unlabeled test image(s) given an estimated forward model:
> > > > >
> > > > > - @Vasilescu2011, author={M. A. O. Vasilescu}, booktitle={Proc. IEEE Inter. Conf. on Automatic Face Gesture Recognition (FG 2011)}, title={Multilinear projection for face recognition via canonical decomposition}, year={2011}, pages={476-483}, doi={10.1109/FG.2011.5771445}, month={Mar}}
> > > > >
> > > > >Tensor causal factor analysis paper that split a data tensor into
> > > > > data blocks and specifies which data blocks share the same mode matrix and hence contribute to the mode matrix computation.
> > > > > - @inproceedings{Vasilescu20,
> > > >     author={Vasilescu, M. Alex O. and Kim, Eric and Zeng, Xiao S.},
> > > >     booktitle={2020 25th International Conference of Pattern Recognition (ICPR 2020)},
> > > >     title={Causal{X}: Causal e{X}planations and Block Multilinear Factor Analysis},
> > > >     year={2021},
> > > >     location={Milan, Italy},
> > > >     month={Jan},
> > > >     pages={10736--10743},
> > > >     }}}
> > > >
> > > > Although our paper has no direct connection to causal factor analysis, we have cited them in Introduction to emphasis differences from existing methods.
> > > > We do not cite `Panagakis2021` and `Kossaifi20` as we do not study regression. We have cited `levoy1996light` and `gortler1996lumigraph` in Section 3. Our model is nothing to do with Tensor Texture model, then we do not cite `Vasilescu04`.
> > > >
> > > > --
> > > >
> > > > We appreciate a lot for your contribution of reviewing. We acknowledge that you are continuously revising your review. To avoid misunderstanding between us, it would be great if you could leave review history so that we (and other reviewers and area chairs) can see what has been changed.

---

### Decision · Program_Chairs · 2023-01-20

**Decision:**

Reject

**Justification For Why Not Higher Score:**

Although the paper has received some higher scores, based on my own reading I can not recommend acceptance. I think, that the numerical experiments should be stronger, and the novelty is limited (decomposition is known before as well as the algorithms). No non-trivial algorithms for selecting the set for $B$ is given, so when writing this lines I am quite confused by the nice evaluation by two of the reviewers. I think the algorithm has potential, but it has to be tested and evaluated at a different level.

**Justification For Why Not Lower Score:**

N/A

**Metareview: Summary, Strengths And Weaknesses:**

The paper discusses and proposes a tensor decomposition based on the Legendre decomposition of non-negative tensors which represent probability distribution. The decomposition is parametrized in terms of the $\theta$-parameters, which are essentially the logarithm of the probability up to a certain triangular-type linear transformation.

Strengths: Interesting observation is that the optimization problem (once the set of non-zero indices is fixed) can be written as a convex optimization, which is not true, for example, for other tensor decompositions.

Weaknesses. The representation is a product representation, without any sum. So, after taking the logarithm, we get a linear model with some non-negativity constraints (but I admire that the model itself is interesting, but seems it has been proposed before).

There are also some comments about connection to tensor ring decomposition. The properties of the tensor-ring decomposition are completely different from the tensor-train: since the tensor network for the tensor ring decomposition has cycles, this format is in general unstable, whereas tensor-train is stable. The tensor ring for particular tensors can give better compression.

Overall, I don't see which representation should be more stable, it probably will depend on the application.  The connection to tensor networks and the expressive power is not given, as well as more illustrative and practical experiments.


Numerical examples are on very strange and non-standard datasets, which can not be considered characteristics. One can take the standard functions, such as $\sin(x_1 + \ldots + x_d)$ or $\frac{1}{x_1 + \ldots + x_d}$ for testing the accuracy of different decompositions, rather than obscure example with no clear answers.